# A conserved population of MHC II-restricted, innate-like, commensal-reactive T cells in the gut of humans and mice

Carl-Philipp Hackstein [1,2], Dana Costigan[3], Linnea Drexhage [2,9], Claire Pearson [4], Samuel Bullers[4], Nicholas Ilott [4], Hossain Delowar Akther[1,2], Yisu Gu[4], Michael E. B. FitzPatrick [2], Oliver J. Harrison [5,6], Lucy C. Garner [2], Elizabeth H. Mann [4], Sumeet Pandey[2], Matthias Friedrich[2,4], Nicholas M. Provine [2], Holm H. Uhlig [7], Emanuele Marchi[2], Fiona Powrie [4], Paul Klenerman [1,2,10] ✉ & Emily E. Thornton [3,4,8,10] ✉

Interactions with commensal microbes shape host immunity on multiple levels and play a pivotal role in human health and disease. Tissue-dwelling, antigen-specific T cells are poised to respond to local insults, making their phenotype important in the relationship between host and microbes. Here we show that MHC-II restricted, commensal-reactive T cells in the colon of both humans and mice acquire transcriptional and functional characteristics associated with innate-like T cells. This cell population is abundant and conserved in the human and murine colon and endowed with polyfunctional effector properties spanning classic Th1- and Th17-cytokines, cytotoxic molecules, and regulators of epithelial homeostasis. T cells with this phenotype are increased in ulcerative colitis patients, and their presence aggravates pathology in dextran sodium sulphate-treated mice, pointing towards a pathogenic role in colitis. Our findings add to the expanding spectrum of innate-like immune cells positioned at the frontline of intestinal immune surveillance, capable of acting as sentinels of microbes and the local cytokine milieu.

The mucosal surfaces of the intestine are colonized by myriad commensal microbes that outnumber the host's own cells and play crucial roles in metabolizing certain nutrients as well as limiting the growth of potentially pathogenic microbes[1,2]. In this unique environment, a delicate balance is required to restrict antimicrobial immune responses to a minimum necessary to contain the microbes within the intestinal lumen while avoiding excessive inflammatory processes that would damage the host tissue[3]. MHC II-restricted CD4+ T cells that are responsive to microbial antigens are a major immune cell population in the intestine and were shown to play key roles in both homeostasis and chronic inflammation[4]. Considerable efforts have been made in the past to unravel how interactions with intestinal microbes shape the phenotype and function of these immune cells, including experiments in gnotobiotic mice[5–7], tetramer-based approaches analyzing antigen-specific endogenous cells[8–10], and experiments utilizing TCR transgenic mice to study the evolution of the response to microbial

[1]Peter Medawar Building for Pathogen Research, University of Oxford, Oxford, UK. [2]Translational Gastroenterology Unit, Nuffield Department of Medicine, University of Oxford, Oxford, UK. [3]MRC Human Immunology Unit, MRC Weatherall Institute of Molecular Medicine, University of Oxford, Oxford, UK. [4]Kennedy Institute of Rheumatology, NDORMS, University of Oxford, Oxford, UK. [5]Center for Fundamental Immunology, Benaroya Research Institute, 1201 9th Ave, Seattle, WA 98101, USA. [6]Department of Immunology, University of Washington, 750 Republican St, Seattle, WA 98108, USA. [7]Translational Gastroenterology Unit, and Biomedical Research Centre, and Department of Paediatrics, University of Oxford, Oxford OX39DU, UK. [8]Nuffield Department of Medicine, University of Oxford, Oxford, UK. [9]Present address: Sir William Dunn School of Pathology, University of Oxford, Oxford, UK. [10]These authors jointly supervised this work: Paul Klenerman, Emily E. Thornton. ✉e-mail: paul.klenerman@ndm.ox.ac.uk; emily.thornton@imm.ox.ac.uk

antigens[8,11–15]. Using these tools, previous studies demonstrated that the type of antigen, as well as the milieu during activation, has a huge impact on local T cell differentiation, which results in a diverse continuum of phenotypes spanning from regulatory T cells (Treg)[11,16] to Th1 and Th17 cells[8,10].

Cytokine-mediated signals play a crucial role in inducing, regulating, and maintaining T cells in tissues[17–20]. IL-18R1 expression is more frequent on lamina propria CD4 T cells compared to their counterparts in lymphoid tissues[21], and signaling mediated by IL-18, IL-12, and type I interferon plays important roles in shaping T cell phenotypes and functions in the intestine[21,22]. This cytokine responsiveness is a feature conventional MHC-restricted T cells share with other immune cells enriched in mucosal sites, including innate-like T cells, which can also recognize bacterial products through their TCRs. Interestingly, it also has been shown that commensal bacteria regulate the expansion of a CD4 T cell subset with a diverse TCR repertoire and innate-like characteristics including PLZF expression in transgenic mice[23]. Further, a notable PLZF+ subset can be identified in the intestinal CD4 T cells in human fetal tissues[24], and in adults, many human colonic CD4 T cells express the C-type lectin CD161[25], a key marker of human innate-like T cells such as the semi-invariant Vδ2 population[26,27] as well as with iNKT and MAIT cells[28]. The association between high CD161 expression and the ability of T cells to respond to cytokine stimulation in a TCR-independent, innate-like manner is not restricted to these well-defined subsets and can been observed in small populations across all T cell lineages[29]. Such a capacity for cytokine-responsiveness in an environment as rich in stimulating agents as the gut could have important consequences in both health and disease.

In this study, we seek to understand whether microbe-reactive cells in the gut may be able to span the MHCII-restricted conventional T cell population and cytokine-responsive innate-like T cells. We describe that in the human colon, microbe-reactive CD4 T cells have high CD161 expression, which is associated with the acquisition of key features of innate-like T cells including PLZF expression and the ability to respond to cytokines including IL-12, IL-18, and IL-23 independently of TCR-signals even though they are MHCII-restricted and express a diverse TCR repertoire. CD161$^{hi}$ CD4 T cells accounted for the vast majority of CD4 T cells responding to several common commensal microbes placing these cells at the forefront of antimicrobial responses during homeostasis and inflammation. Interestingly, the transcriptional signature defining these MHC II-restricted, innate-like, and commensal reactive T cells (T$_{MIC}$) did not strongly correlate with established T$_{RM}$ signatures, suggesting a unique differentiation pathway. In mice, the transcriptional and functional characteristics of human T$_{MIC}$ cells are mirrored by a population of MHC II-restricted CD4/CD8 double-negative alpha/beta T cells. Strikingly, the comparison of the commensal-reactive Cbir-TCR transgenic T cells with a helicobacter-reactive TCR transgenic, whose target is absent from the flora, revealed that only the commensal-reactive cells acquire a T$_{MIC}$ phenotype in the murine gut, demonstrating its dependency on microbial antigen availability. Both human and murine T$_{MIC}$ cells displayed a mixed Th1/Th17 effector profile, and T$_{MIC}$ cells contributed to pathology in a murine colitis model.

## Results

### Human microbe-reactive CD4 T cells express high levels of CD161

The gut contains a broad range of different microbe-reactive CD4 T cells, which produce TNF when exposed to their respective microbe in vitro[30] (Fig. 1A, B). As we intended to focus on conventional CD4 T cells, we excluded cells expressing TCR chains associated with known unconventional T cells populations and focussed on CD4 T cells negative for Vα7.2 (expressed by MAIT and GEM T cells), Vα24-Jα18 (expressed by iNKT cells) and TCRγδ (Supplemental Fig. 1A). Studying

the commensal-reactive human colonic CD4 T cells responding to *E. coli, S. aureus,* and *C. albicans*, we noticed that the majority of TNF-positive CD4 cells expressed above-average levels of the C-type lectin CD161 (Fig. 1A). While CD161-expressing colonic CD4 T cells as a whole were described as a population of cells with Th17 characteristics before[25], we were intrigued by the fact that antimicrobial responses seemed to be enriched within cells expressing the highest level of this marker and therefore wanted to study the role of CD161-expressing CD4 T cells in the gut in more detail. To that end, we subdivided the colonic CD4 population into CD161-negative, CD161$^{int}$ and CD161$^{hi}$ cells, defining CD161$^{hi}$ based on CD161 expression levels observed on a discrete subset of CD4 T cells co-expressing high levels of CD161 with another NK-cell marker, CD56 (Fig. 1C). The rationale behind this gating strategy was that CD56 expression can be used to pull out a subset of MAIT cell with higher sensitivity towards cytokine stimulation from the larger CD8 T cell population[31]. As CD56 expression is not a uniform feature of the entire MAIT population, we decided to study both double-positive (DP, CD161$^{hi}$CD56+) and CD161$^{hi}$ CD56- CD4 T cells in most assays. TNF-expressing microbe-responsive cells were found among both CD161$^{hi}$CD56- and DP CD4 T cells, and the frequency of TNF-positive, microbe-responsive cells was significantly higher in both populations compared to either CD161$^-$ or CD161$^{int}$ cells. (Fig. 1D). Compared to each other, there was a non-significant trend for higher percentages of TNF-positive cells within the DP population.

In sum, our data show that the vast majority of human colonic microbe-responsive CD4 T cells display a CD161$^{hi}$ phenotype with varying expression of CD56.

### MHC-II-restricted CD161$^{hi}$ colonic CD4 T cells share transcriptional and functional characteristics with innate-like T cells

Further phenotypic profiling revealed that almost all microbe-reactive CD161$^{hi}$ CD4 T cells expressed IL18Rα, while expression of IFNγ varied between the different microbe-specific populations (Fig. 1E, F). Strikingly, our experiments also revealed that PLZF, the key transcription factor regulating the development and function of innate-like T cells, was expressed in microbe-responsive cells at higher levels compared to the bulk of non-responding TNF- cells and was also more highly expressed in microbe-reactive cells compared to CD4 T cells producing TNF in response to staphylococcal enterotoxin B (SEB, Fig. 1G, H).

These findings prompted us to test whether these antimicrobial responses were dependent on the conventional CD4 T cell restriction element MHC II or if there was any role for unconventional antigen-presenting molecules like MR1 and CD1d. MHC II blocking massively reduced or abrogated the microbe-induced TNF production by human colonic CD4 T cells (Fig. 1I, J), supporting previous work showing these commensal-reactive cells are MHC-II restricted[30]. In contrast, blocking of CD1d and MR1, the molecules presenting antigens to type I and II NKT cells and MAIT cells respectively, had no significant effects on antimicrobial responses. Similar, the addition of neutralizing antibodies against IL-12 and IL-18 failed to block TNF production (Fig. 1I, J).

Taken together, this indicates that commensal-responsive CD161$^{hi}$ CD4 T cells, despite expression of several markers associated with innate-like T cells (CD161, higher levels of cytokine receptors, PLZF) are a population of MHC-II-restricted CD4 T cells.

An intriguing aspect of our findings is that this innate-like phenotype was found in cell populations responding to microbes that, while all being considered commensals in the human intestine, differ notably from each other and included a gram-negative proteobacterium (*E. coli*), a member of the gram-positive firmicutes phylum (*S. aureus*) and the yeast *C. albicans*. As only a small fraction of the entire CD161$^{hi}$ subset responded to each of these microbes (Fig. 1A, D), we hypothesized that the CD161$^{hi}$ population would contain further cell populations with the same phenotype, presumably responsive to microbes not tested in this study.

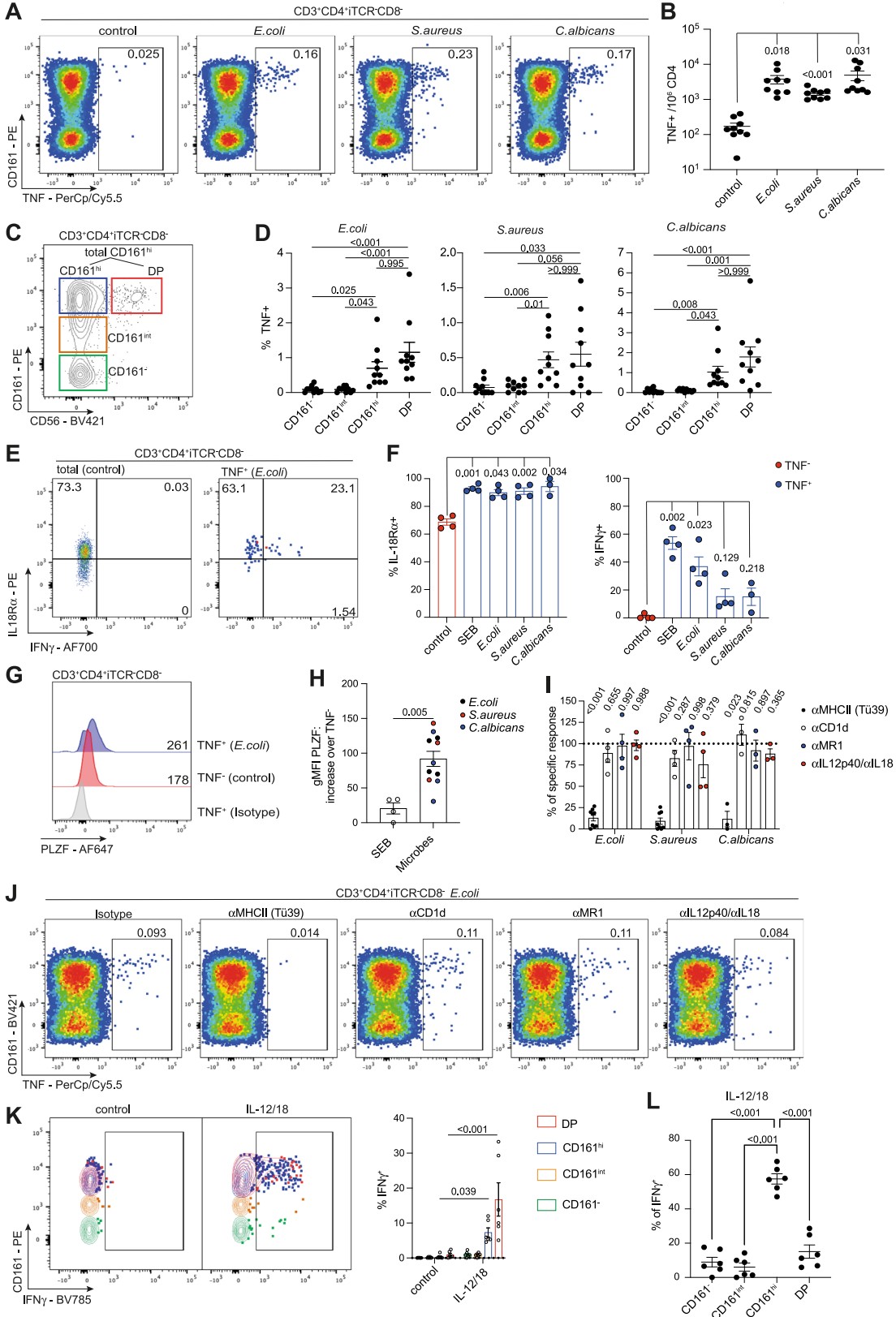

To test this idea, we phenotyped bulk CD161hi CD4 as defined earlier (Fig. 1C) in more detail. Like the microbe-specific subsets, bulk CD161hi CD4s expressed higher levels of PLZF (Supplemental Fig. 1B, C) and showed higher expression of IL18Rα compared to their intermediate and negative counterparts (Supplemental Fig. 1D). Strikingly, RNAseq of human intestinal CD4 T cells showed

that a transcriptional core signature (Supplementary Data 1) previously identified in MAIT cells as well as other innate-like T-cell populations in human blood[29], was strongly enriched in CD161hi CD4 T cells as well (Supplemental Fig. 1F), suggesting that these CD4s indeed share transcriptional features with established innate-like T cell subsets.

**Fig. 1 | Human microbe-reactive CD4 T cells are CD161$^{hi}$ and display a poly-functional effector phenotype with innate-like features.** See also Supplemental Figs. 1–3. **A** Expression of TNF by iTCR (Vα7.2, Vα24-Jα18, TCRγδ)-negative colonic CD4 T cells after 8 h of incubation in medium (control) or heat-killed microbes. **B** Absolute numbers of TNF-positive CD4 T cells per million colonic CD4 T cells, n = 9 biologically independent samples. **C** Expression of CD161 and CD56 by colonic CD4 T cells after exclusion of cells expressing TCRs associated with innate-like T cells (iTCR: see **A**) or CD8. **D** Frequencies of TNF-expressing CD161$^-$, CD161$^{int}$, CD161$^{hi}$, and double-positive (DP) CD4 T cells after 8 h of microbial stimulation, n = 10 biologically independent samples. **E** Expression of IL18Rα and IFNγ on TNF$^-$ or TNF$^+$ CD4 T cells upon stimulation with *E. coli*. **F** Percentage of IL18Rα and IFNγ-positive cells among TNF$^-$ or TNF$^+$ CD4 T cells upon stimulation with SEB or microbes, n = 4 biologically independent samples. **G** Expression of PLZF in TNF$^-$ or TNF$^+$ CD4 T cells upon stimulation with *E. coli*. **H** Difference in PLZF expression between TNF$^-$ CD4 T cells compared to TNF$^+$ CD4 T cells, n = 4 biologically independent samples. **I** Microbe-specific TNF responses upon blockade of MHC-II, CD1d, MR1, or IL12/18 as a percentage of the response observed with isotype antibodies (100%, dotted line). Statistics were calculated comparing the responses for each blocking condition to the normalized response in each donor, n = 3 (*C. albicans* responses), 4 (MR1, CD1d, and IL12/18-blocking), or 10 (MHC-II blocking of *E. coli* or *S. aureus* responses) biologically independent samples. **J** *E. coli*-induced TNF-expression by colonic CD4 T cells in the presence of blocking antibodies or an isotype control. **K** Expression of IFNγ by colonic CD4 T cells after 20 h of stimulation with IL-12/18, n = 6 biologically independent samples. **G** Percentages of IFNγ-positive cells among the populations from **K**. (**B, D, F, H, I, K, L**): data points were pooled from independent experiments using one or two human samples each. Statistics: repeated measures ANOVA with Dunnett's multiple comparisons test (**B, L**), Friedman tests with Dunn's multiple comparisons tests (**D**), Mixed-effects analysis with Tuckey's or Dunnett's multiple comparisons test (**F, I**), two-tailed Mann–Whitney test (**H**), 2-way ANOVA with Sidak's multiple comparisons test (**K**). Mean ± SEM is shown. Source data are provided as a Source Data file.

The ability to mount TCR-independent responses to cytokine stimulation is a key functional property of innate-like T cells and hence, we assessed how the different colonic CD4 T cell subsets responded to IL-12/18 stimulation. Bulk CD161$^{hi}$ CD4 were able to respond to combined IL-12 and IL-18 stimulation in the absence of additional TCR stimulation (Fig. 1K) and accounted for the majority of responding CD4 T cells (Fig. 1L). As reported for CD56+ and CD56- MAIT cells[31], DP CD4 T cells showed a higher percentage of IFNγ-production than their CD161$^{hi}$ (CD56$^-$) counterparts, but overall the responses of both subsets resembled the cytokine-induced, TCR-independent responses of CD161$^{hi}$ innate-like T cells like MAIT, iNKT, and Vδ2 + γδT cells.

Innate-like T cells including γδ T[32–35], iNKT[36,37], MAIT[38–40], and H2-M3-restricted T cells[41] have been shown to be involved in tissue repair in different barrier organs. Having established that human CD4 CD161$^{hi}$ T cells harbor populations of microbe-reactive T cells with innate-like characteristics localized in the colonic lamina propria, we also assessed whether they would also express tissue repair-associated factors upon stimulation. The regulation of tissue repair is a complex process involving many different effector molecules and pathways. Hence, we used a multi-modal single-cell sequencing approach to analyze the expression of an established gene list of tissue repair-associated factors[39,41] in colonic CD4 T cells that had been stimulated with either cytokines (IL-12 and IL-18), plate-bound anti-CD3 antibodies or a combination of both overnight (Supplemental Fig. 2). Gene counts of repair-associated factors were barely affected in CD161$^{hi}$ CD4 T cells that had received cytokine stimulation (Supplemental Fig. 2A), while TCR or combined stimulation induced increased expression compared to unstimulated control cells (Supplemental Fig. 2B, C). In line with that, gene set enrichment analyses (GSEA) showed that, indeed, the tissue repair gene set was enriched in TCR and TCR+ cytokine-stimulated, but not cytokine-only stimulated CD161$^{hi}$ CD4 T cells (Supplemental Fig. 2D). Leading edge genes driving the enrichment included CSF1 and 2, the genes encoding m-CSF and GM-CSF, growth factors like VEGFA and AREG, the gene encoding Amphiregulin (Supplemental Fig. 2E). To conclude, human CD161hi CD4 T cells show the same TCR-dependent potential for the production of tissue repair-associated factors previously found in innate-like T cells.

TCR-sequencing in three donors revealed that bulk CD161$^{hi}$ CD4 T cells use a diverse range of private TCRα and -β chains in line with the idea that this population consists of a pool of MHC II-restricted cells responding to a vast range of different microbes. Interestingly, while TCRα diversity was comparable to the repertoire diversity found in CD161$^-$ CD4 T cells in both CD161$^{hi}$ subsets regardless of CD56 expression, DP CD4 T cells showed some bias in their TCRβ usage (Supplemental Fig. 2F). However, no public TCRβ sequences could be found, suggesting that the DP phenotype can arise in CD161$^{hi}$ CD4 T cells with different TCRs and this population might be the result of private clonal expansion.

Additional phenotyping revealed enrichment of RoRγt−expressing cells among CD161$^{hi}$ CD4 T cells (Supplemental Fig. 3A, B). To explore possible Th17-functionality, we stimulated CD161$^{hi}$ CD4 T cells with IL-23, a cytokine playing a pivotal role in gut-associated inflammatory diseases[42–45]. Interestingly, IL-23 alone induced the production of IFNγ and GzmB by a small subset of cells in a TCR-independent manner (Supplemental Fig. 3C, D). CD161$^{hi}$ CD4 T cells could also produce the Th17 cytokines IL-17A, IL-17F, and IL-22 to a limited extent (Supplemental Fig. 3D). The production of the latter cytokines however was strictly dependent on an additional TCR-stimulus.

## Innate-like MHC-II-restricted CD161$^{hi}$ colonic CD4 T cells display an effector memory phenotype and express a transcriptional program different from classic T$_{RM}$ cells

Flow cytometric analyses showed that the majority of CD161$^{hi}$ CD4 T cells displayed an effector memory phenotype (Tem, Supplemental Fig. 3E), in contrast to CD161$^-$ and CD161$^{int}$ CD4 T cells that also harbor major Tcm and naïve populations. Given the tissue-origin and characteristics displayed by CD161$^{hi}$ CD4 T cells, we wondered if their transcriptional phenotype would correlate with the acquisition of genetic signatures associated with tissue-resident memory T cells (T$_{RM}$). To explore this, we obtained a list of CD161$^{hi}$ signature genes by comparing gene expression in CD161$^{hi}$ and DP CD4 T cells with CD161$^-$ CD4 T cells. Between the genes that were significantly upregulated (log2 foldchange >2, adjusted *p* value <0.05, Supplementary Data 2) in CD161$^{hi}$ and DP CD4 T cells there was an overlap of 26 genes, including KLRB1 (CD161), IL23R, IFNGR1, GPR65 and the transcription factors RORA and BHLHE40. Next, we assessed the expression of two previously published T$_{RM}$ gene signatures[46,47] in the CD4 cells from two publicly available intestinal single-cell datasets[48,49]. We then compared the expression of both T$_{RM}$ signatures with the corresponding expression of our CD161$^{hi}$ signature gene module on the single-cell level. Expression of the two different T$_{RM}$ signatures showed a strong positive correlation in both datasets, as expected (Supplemental Fig. 3F). On the other hand, the CD161$^{hi}$ signature module correlated only weakly with both T$_{RM}$ signatures (Supplemental Fig. 3G, H). Based on this, we concluded that the acquisition of the effector profile and innate-like features in CD161$^{hi}$ CD4 T cells represents a unique feature of microbe-reactive CD4 T cells and not an inherent component of the T$_{RM}$ - associated gene signatures in the colon.

Taken together our data suggested the potential for microbe-reactive T cells to take on an innate-like phenotype with broad cytokine responsiveness in the gut. However, since experiments with ex vivo human cells are limited, we decided to determine whether an equivalent subset of MHC-II restricted, innate-like, commensal-reactive T cells (T$_{MIC}$) exist in mice to allow more extensive mechanistic in vivo experimentation.

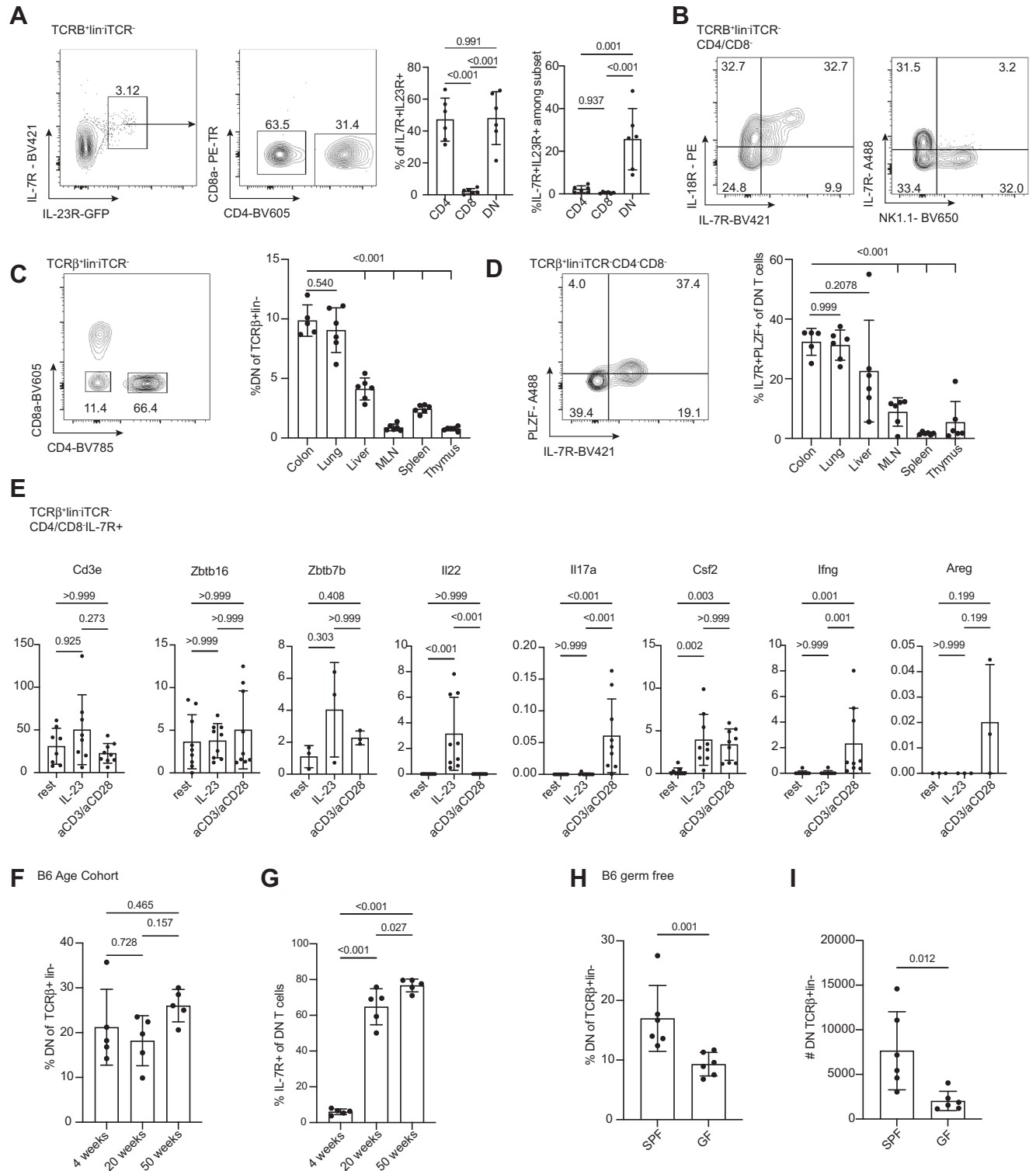

## MHC-II restricted cytokine and microbe-reactive T cells in mice are CD4/CD8 double-negative

Given the innate-like phenotype of the microbe-reactive T cells found in human colon and the absence of an easily identifiable mouse ortholog for CD161, we used IL-23R reporter mice to identify a population of cytokine-responsive conventional (TCRalpha/beta+, not staining positive for CD1d or MR1 tetramers) T cells in the murine colonic lamina propria (Fig. 2A). The IL23R-reporter signal was associated with expression of IL7R, which was subsequently used as a surrogate marker in wildtype animals. When we looked further to

define this IL23R+ IL7R+ subset, which should contain Th17 cells, we found that a majority did not express either CD4 or CD8a on the surface (Fig. 2A), similar to cells described in the peritoneum of the IL-23R reporter[50]. CD4/CD8 double negative (DN) cells have been described in TCR transgenic lines[51] and settings of chronic inflammation such as lupus[52] and spondyloarthropathy[53], but their function has remained enigmatic. The subset of DN cells in the colon that expressed IL-7R also expressed IL-18 receptor but not NK1.1, indicating a population that is broadly cytokine-responsive (Fig. 2B). Gene expression analysis and intracellular staining demonstrated that the absence of CD4 and CD8

**Fig. 2 | Mouse cytokine-responsive CD4−/CD8− mucosal T cells respond to microbial stimuli.** See also Supplemental Fig. 4. Mouse immune cells were isolated from various tissues and analyzed by flow cytometry and/or stimulated in vitro. **A** Flow cytometric analysis of colonic lamina propria leukocytes from IL-23RGFP/+ mice gated TCRβ + lin-iTCR- (MR1tet-CD1dtet-TCRγδ-CD11c-CD11b-B220-) showing IL-7R and IL-23R co-expressing6 cells (left). The gated IL-7R+/IL-23R+ population is shown for CD4 and CD8a expression followed by quantification of the percent of the IL7R+/IL23R+ population is either CD4+, CD8a+ or CD4/CD8a double negative (DN). Quantification of gated TCRβ+ in- CD4, CD8a, or DN cells that have the IL-7R/IL-23R positive phenotype. Each point represents one individual mouse (n = 6), representative of three independent experiments. One-way ANOVA with Tukey's multiple comparison test. **B** Example flow cytometry plot shows TCRβ + lin-iTCR-CD4/CD8− cells expressing IL-7R also express the IL-18R (left). IL-7R and NK1.1 are not co-expressed in DN T cells (right). Plots are from different experiments but representative of two experiments each, n = 5. **C** Example FACS plot (colon) and quantification of DN phenotype in mouse colon, lung, liver, mesenteric lymph node (MLN), spleen, and thymus. Each point represents one individual mouse. n = 5 for colon, n = 6 for all others, representative of two independent experiments. One-way ANOVA with Dunnett's multiple comparisons test comparing each mean to colon. **D** Example FACS plot (colon) and quantification of T cells innate-like phenotype (IL-7R+/PLZF+) in mouse colon, lung, liver, MLN, spleen, and thymus. Each point represents one individual mouse. n = 5 for colon, n = 6 for all others, representative of two independent experiments. One-way ANOVA with Dunnett's multiple comparisons tests comparing each mean to colon. **E** FACS sorted TCRβ + lin-iTCR-CD4/CD8− cells were isolated from mouse colon and stimulated in vitro with either aCD3/aCD28 or IL-23 or rested in complete media for 24 h when RNA was extracted and qPCRs performed for *Cd3e*, *Zbtb16* (PLZF), *Zbtb7b* (ThPOK), *Il22*, *Il17a*, *Csf2* (GM-CSF), *Ifng*, and *Areg*. Genes were normalized to *Hprt*. Each point represents one mouse. Data represent two combined experiments, n = 9 in total except for *Zbtb7b* and *Areg* n = 3. Kruskal–Wallis with a Dunn multiple comparison test. **F** Quantification of FACS data from DN TCRβ+lin− cells from colon LPLs from mice aged 4, 20, and 50 weeks. n = 5, representative of two independent experiments. One-way ANOVA with Tukey's multiple comparison test. **G** Quantification of FACS data of IL-7R expression on the DN T-cell population of 4, 20, and 50-week-old mice. Each point represents an individual mouse. n = 5, representative of two independent experiments. Mean with standard deviation shown, one-way ANOVA with Tukey's multiple comparison test. **H** Quantification of FACS data of DN TCRβ + lin− cells from colon LPLs from specific pathogen-free (SPF) or germ-free (GF) mice. Each point represents an individual mouse. n = 6, representative of three independent experiments. Unpaired t test. **I** Absolute cell numbers of the colonic DN cell population of SPF and GF mice. Each point represents and individual mouse. n = 6, representative of three independent experiments with unpaired t test. Mean with standard deviation shown for all graphs. Source data are provided as a Source Data file.

was not due to collagenase digestion or internalization (Supplemental Fig. 4A, B). While we used MHC-II blocking antibody to demonstrate MHC-II restriction of the innate-like cells from human tissue, we were able to use the B2M knockout mouse to eliminate the possibility that we were studying a known innate-like T cell population. While the B2M knockout showed a slight decrease in DN cell numbers, the overall phenotype of the cells was similar (Supplemental Fig. 4C), demonstrating that these cells are not dependent on MHC-I, CD1d, or MR1. To determine whether the presence of this DN T cell population was specific to the gut, we examined mucosal and non-mucosal tissue sites. While lymphoid organs were almost devoid of DN T cells, the barrier sites contained abundant DN T cells (Fig. 2C) with the lungs containing a similarly sizable population of innate-like T cells (Fig. 2D).

To determine whether these cells were phenotypically and functionally similar to human cytokine-responsive cells, we performed flow cytometry and in vitro stimulations. As with the human cells, mouse colonic DN T cells contain a PLZF-expressing population (Supplemental Fig. 4D). As expected, these cells expressed CD3e, Zbtb7b (ThPOK), a transcription factor associated with Th cells, and Zbtb16 (PLZF), which do not change upon TCR or IL-23 stimulation. By contrast, IL-22 production was stimulated by IL-23 treatment while IL-17A and IFNg production were stimulated by TCR stimulation. GM-CSF was produced in both conditions. These data suggest that the DN T cell population is similar to the cytokine and microbe-responsive population found in the human colon and has the ability to tune their response based on the cytokine milieu and/or antigen stimulation (Fig. 2E).

In order to determine whether these cells are likely to be the microbe-responsive T cells of interest in the murine gut, we examined mice from different ages, supposing that older mice would have encountered more microbes and mild insults that lead to microbe translocation into the lamina propria. While the percent of the overall T cell population with the CD4/CD8 DN phenotype did not change with age (Fig. 2F), the proportion with a cytokine-responsive phenotype, marked by IL-7R, increased markedly between 4 and 20 weeks of age and still more by 50 weeks of age (Fig. 2G). Conversely, germ-free mice, devoid of microbes, had lower percentages and absolute numbers of DN T cells (Fig. 2H, I).

## MHC-II-restricted microbe-reactive cells develop an innate-like phenotype in the gut

In order to explore further the function of gut-associated microbe-reactive T cells and to determine whether the innate-like phenotype can be observed in a known MHC-II restricted, microbe-reactive T

cell, we turned to the Cbir1 TCR transgenic mouse[12]. The Cbir1 transgenic T cell recognizes a flagellar antigen from Clostridium cluster XIVa, and when crossed onto the Rag1 knockout background (Rag−/−), the transgenic mouse only contains T cells of one specificity without the possibility of recombination. Hence, these mice represent an ideal model to study the phenotype and functions of commensal-reactive MHC-II-restricted T cells in vivo. As observed in B6 mouse colons, the colon of the Cbir1 mouse contained a population of CD4/CD8 negative cells (Fig. 3A, B), but in the context of the TCR transgenic, approximately 70% of the T cells had this phenotype. As seen with the other microbe-reactive cells from human and mouse, these cells expressed IL-18 R and IL-7 R (Fig. 3A, B), suggesting they are broadly cytokine-responsive in addition to being microbe-reactive. Unlike T_MIC cells in B6 mice, Cbir1 T cells are specific for a single commensal that is found in the gut. The prevalence of the DN phenotype is generally restricted to the colon in this setting suggesting antigen presence is important for the development and/or retention of these cells (Fig. 3C). Like DN T cells in wildtype animals, Cbir DN T cells also expressed PLZF in the colon (Fig. 3D); however, a proportion of the DN cells in the lung also had PLZF expression suggesting there may be some crosstalk between the tissues. Analysis of the thymus from B6 and Cbir1 mice demonstrated a very small but appreciable population of T_MIC cells in the thymus (Supplemental Fig. 3E) suggesting that cells may develop in the thymus similar to other innate-like T cells and populate mucosal sites where they are able to persist. By contrast, another MHC-II-restricted TCR transgenic mouse strain *(Hh-TCR Rag−/−)* specific for a microbe, *Helicobacter hepaticus*, whose antigen is absent from our animal facility did not contain many DN T cells (Supplemental Fig. 4E, F), suggesting that antigen stimulation is likely required for development or retention in the gut. To verify that DN T cells still responded to their antigen in a TCR/MHC-II-dependent manner, cells were stimulated in vitro with peptide-loaded antigen-presenting cells (APCs). Sorted DN Cbir1 transgenic T cells indeed responded to their cognate peptide, and their proliferation could be blocked with an MHC-II blocking antibody (Fig. 3E). To further demonstrate that proliferation to the native antigen was MHC-II dependent, APCs were fed feces from steady-state or vancomycin treated mice overnight before T cell introduction, and proliferation was again blocked by MHC-II blocking antibodies (Fig. 3F). This demonstrated that our animal facility contained the antigen the T cells respond to and that proliferation was MHC-II dependent. Taken together, these data suggest T_MIC cells can develop in the mouse.

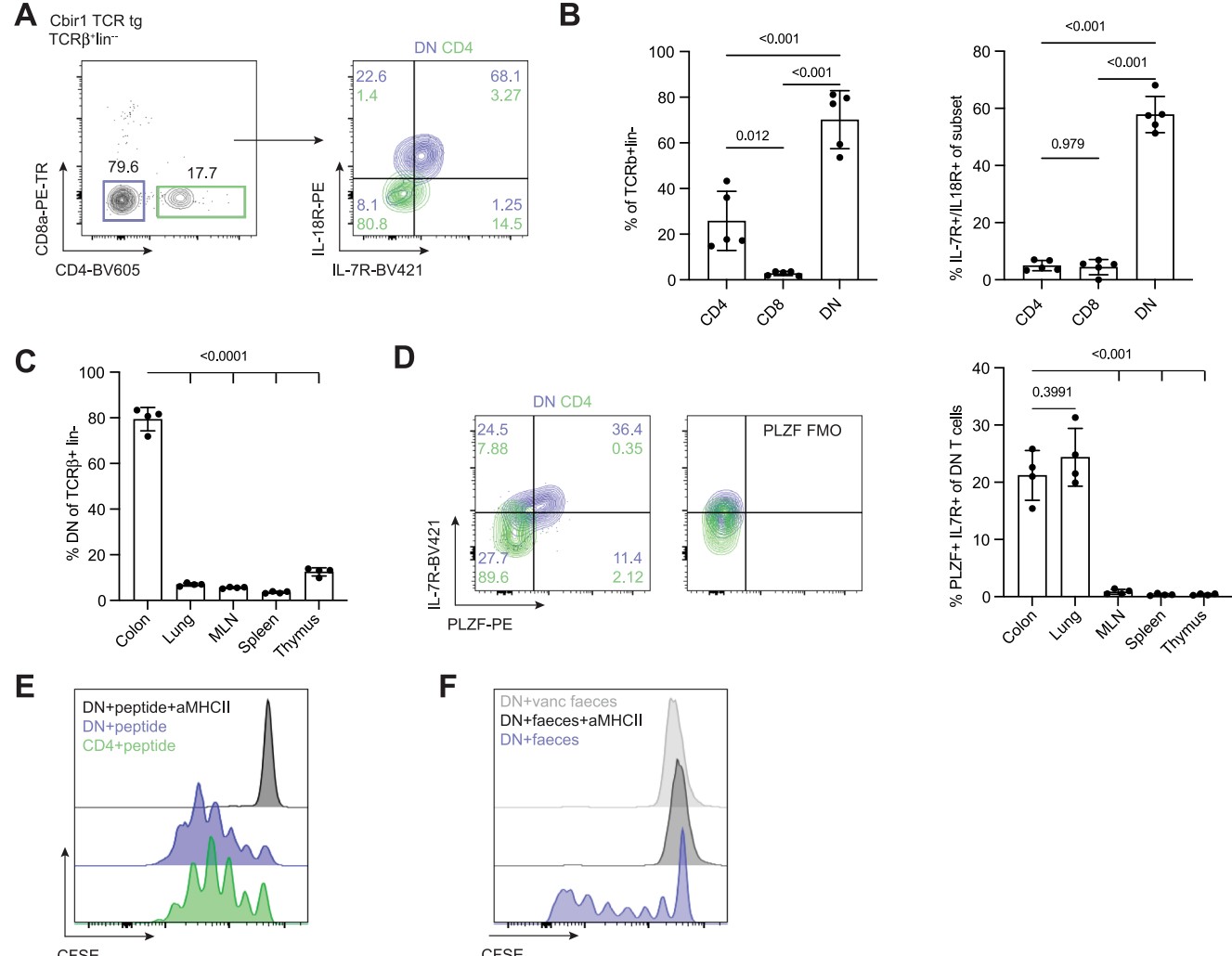

**Fig. 3 | Cbir1, MHC-II restricted, microbe-reactive TCR transgenic develop an innate-like phenotype in the gut.** LPLs from Cbir1 TCR transgenic line crossed to Rag−/− mice were analyzed directly ex vivo or stimulated in vitro. **A** Flow cytometric analysis of TCRβ + lin− cells from the colon lamina propria of Cbir1 Rag−/− mice stained for CD4 and CD8a. (left) and IL-7R and IL-18R stain of the gated CD4 (green) and CD4/CD8 double negative (DN, blue) cell populations (right). FACS plots are representative of experiments n = 5, representative of two independent experiments. **B** Quantification of flow cytometry data showing the percent of the TCRβ + lin− population that are CD4, CD8a, and DN (left), and the percent of each population that have the IL-18R/IL-7R expressing phenotype (right). Each point represents an individual animal. n = 5, representative of two independent experiments. Mean with standard deviation shown, one-way ANOVA with Tukey's multiple comparison test. **C** Quantification of FACS data of the percent of T cells with a DN phenotype from Cbir1 colon LPL, lung, mesenteric lymph node (MLN), spleen, and

thymus. Each point represents an individual mouse. n = 4, representative of two independent experiments. Mean with standard deviation shown, one-way ANOVA with Dunnett's multiple comparisons test comparing each mean to colon.
**D** Representative flow cytometry data (left) and quantification (right) of IL-7R and PLZF expression within the CD4 and DN T cell subsets with fluorescence minus one (FMO) control (right). **E** Representative flow cytometry histograms of sorted CD4 or DN splenic Cbir1 Rag−/− T cells labeled with CFSE and incubated for 5 days with peptide-pulsed bone marrow-derived DCs (BMDCs). Plots are representative of three independent experiments. **F** Representative flow cytometry histograms of sorted DN T cells from Cbir Rag−/− mice stained with CFSE and incubated for five days with BMDCs fed feces of steady-state or vancomycin (vanc) treated mice from the same animal facility. Plots are representative of two independent experiments. Source data are provided as a Source Data file.

## Mouse and human T$_{MIC}$ cells share a transcriptional program

In order to determine how similar human and murine T$_{MIC}$ cells are, we performed RNA-sequencing on sorted intestinal T cell populations of interest and several control populations. We merged human and murine RNA-sequencing datasets based on common gene ids as described before[39,54]. This approach allowed us to initially focus on genes present in both datasets, enabling us to identify cell populations with similar transcriptional features based on clustering and principal component analyses (PCA). To that end, we merged a human dataset containing four CD4 T cell populations differing in their expression of CD161 (Supplemental Fig. 5A), a murine dataset containing colonic αβ TCR+ CD4 and double-negative T cells (Supplemental Fig. 5B) as well as iNKTs and a selection of Immgen-derived

T cell datasets[55] as comparators. Strikingly, CD161$^{int}$ and CD161$^{−}$ human CD4 T cells clustered together with murine CD4s based on the first two components of our PCA (Fig. 4A) or hierarchical clustering (Supplemental Fig. 5C), while CD161$^{hi}$ CD4s clustered closer to murine iNKTs and DN T cells, confirming the presence of transcriptional features shared with innate-like T cells described before and the existence of a transcriptional program shared between human and murine T$_{MIC}$ cells. Since the main source of variance (PC1) in our merged dataset separated most the Immgen-derived samples from ours, with the exception of samples that had undergone in vitro αCD3 treatment, we wondered if the inclusion of these data might have a confounding impact on the way human and murine colonic T cells clustered amongst each other. Restricting our analysis to

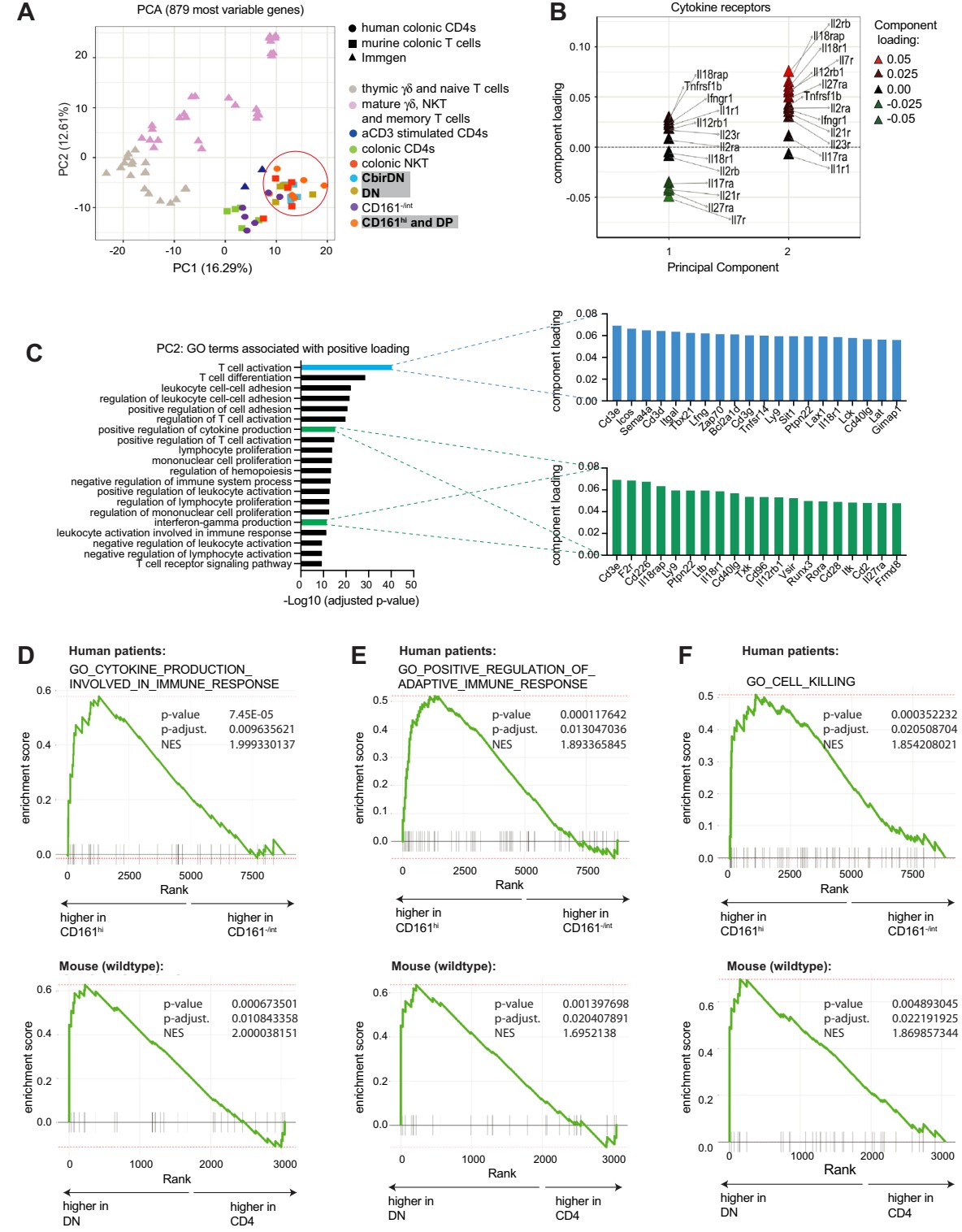

human and murine colonic T cells however lead to similar results, as murine DN T cells and human CD161[hi] CD4 T cells were still split apart from their respective counterparts by PCA and clustered together (Supplemental Fig. 5D).

To explore the shared features of microbe- and cytokine-responsive T cells found in mouse and man, we analyzed which genes were driving the clustering in our PCA (Fig. 4A, Supplemental Fig. 5C, Supplementary Data 3). In line with the fact that PC1 in our merged dataset separated samples originating from spleen, lymph nodes, and the thymus from activated and colon-derived T cells, a GO-

term analysis of the genes positively associated with it indicated that processes linked to activation, differentiation, and adhesion of T cells were the main drivers. Overall, GO: 0042110 "T cell activation" turned out to be the GO-term most significantly associated with PC1 (Supplemental Fig. 5E, Supplementary Data 4). On the other hand, across the entire merged dataset PC2 separated more mature, effector-like T cell types (iNKTs, γδT cells, CD161[hi] CD4 T cells, and DN) cells from others. Interestingly, a PCA-loading analysis revealed that most cytokine-receptor genes contained in the dataset were positively associated with this PC (Fig. 4B), providing a molecular basis for the

**Fig. 4 | Human and murine T$_{MIC}$ cells share transcriptional features and show enrichment of effector-related genes.** See also Supplemental Fig. 5 and Supplemental Fig. 6. Colonic human CD161$^{hi}$ and murine DN T cells as well as several control populations were sorted and subjected to bulk RNA sequencing. The depicted analyses were performed after filtering for non- and lowly expressed genes. **A** Principal component analysis was performed on the 866 most variable genes (IQR > 0.75); plotted are the scores of the first two principal components (PC). Genes were obtained from a merged dataset consisting of human and murine T-cell populations derived from experiments described in this study or obtained from Immgen as indicated. Human and murine datasets were merged based on orthologue genes. **B** Loading analysis of the principal components from **A**, illustrating the contribution of genes encoding cytokine receptors to the overall variance described by PC1 and 2. **C** Analysis showing the top 20 Gene Ontology (GO) terms enriched in the genes positively contributing to PC2 from **A**. For the highlighted GO terms, the respective top 20 genes from PC2 are shown. **D–F** GSEA plots depicting the enrichment of the three indicated GO terms in human (CD161$^{hi}$ CD4) and murine (DN) T$_{MIC}$ cells compared to their respective control populations. NES = normalized enrichment score. Statistics: GO-term enrichment was tested by a one-tailed version of Fisher's exact test implemented in the clusterprofiler R package and $p$ values were multiple hypothesis testing by the Benjamini–Hochberg method (**C**). GSEA p-values were derived from an adaptive multi-level split Monte Carlo scheme implemented in the fgsea R package and adjusted for multiple hypothesis testing by the Benjamini–Hochberg method (**D–F**). Source data are provided as a Source Data file.

observed cytokine-responsiveness of the human and murine cell subsets towards IL-23, IL-12, and IL-18 and suggesting that it might extend beyond those measured by flow cytometry. This was further corroborated by the fact that 2 of the 20 GO terms (Supplementary Data 4) most significantly associated with the genes driving this PC are linked to cytokine responsiveness and production (GO:0001819 "positive regulation of cytokine production" and 0032609 "interferon-gamma production") and include the genes *Il18rap*, *Il12rb*, *Il18r1* and *Ifngr1* (Fig. 4C). Interestingly, GO: 0042110 "T cell activation" again was the most positively associated term, albeit driven by other genes than for PC1.

An important limitation of the data merging method we employed is that genes not present in all of the individual datasets or genes that do not have a direct ortholog in humans or mice respectively, get removed during the merging process. To test whether the characteristic features we identified in our merged dataset are also present in the complete original datasets we separately compared human CD161$^{hi}$ CD4 T cells to their CD161$^{int/-}$ counterparts and murine wildtype or Cbir DN to murine CD4 T cells. Based on differential gene expression analyses, we obtained lists of genes associated with human CD161$^{hi}$ CD4 T cells (human T$_{MIC}$ gene module, Supplementary Data 2) or DN T cells in mice (murine T$_{MIC}$ gene module, Supplementary Data 5). Expression of these gene modules was tested in the respective other species by GSEA, revealing enrichment of the murine T$_{MIC}$ gene module in human CD161$^{hi}$ CD4 T cells and of the human T$_{MIC}$ gene module in murine wildtype and Cbir DN T cells respectively (Supplemental Fig. 6A, B). Genes driving the enrichment of the T$_{MIC}$ gene modules included the human and murine orthologues of IL23R as well as those of the transcription factors RORA, ID2, and BHLHE40. Other notable genes distinguishing human and murine T$_{MIC}$ cells from CD161$^{int/-}$ or CD4 T cells respectively were the motility factor S100A4, the pH-sensor GPR65, and the GPI transamidase PIGS (Supplemental Fig. 6C, D). Additional GSEAs revealed that human CD161$^{hi}$ CD4 and murine DN T cells show an enrichment in GO terms and genes associated with cytokine production and regulation of the immune response (Fig. 4D, E and Supplemental Fig. 5E). We also found that both cell populations also showed an enrichment of the GO-term 0001906 "cell killing" (Fig. 4F), not identified in our merged dataset. This finding is in line with the capability of CD161$^{hi}$ CD4 T cells to produce GzmB, as seen earlier (Fig. 1E), confirming and expanding the array of potential effector functions that can be attributed to human and murine T$_{MIC}$ cells. In contrast to this, GO:0042110 "T cell activation" did not show significant enrichment in any of the analyzed populations (Supplemental Fig. 6F). This is in line with the fact that it was associated with both PC1 and 2 in our analysis, suggesting that genes associated with this term are to some degree upregulated in colonic T cells in general.

## T$_{MIC}$ cells are present in the gut in human and murine colitis
Given the large range of potential effector functions that human and murine T$_{MIC}$ cells possess, we next wanted to explore their functional impact in a disease setting.

In biopsies obtained from inflamed tissue of ulcerative colitis (UC) patients, we found that CD4 T cells in general increase in numbers. The largest increase was seen for the CD161$^-$ and CD161$^{int}$ subsets, but we also noted that cells with a CD161$^{hi}$ phenotype are present in equal or even slightly elevated numbers per gram of tissue compared to samples obtained from non-inflamed or normal tissue (Fig. 5A). We also observed persistence of this cell type in two mouse models of colitis (Supplemental Fig. 7A), and the analysis of a previously published dataset containing single-cell data from healthy controls and IBD patients[48] showed higher expression of a gene module containing the T$_{MIC}$-associated genes *ZBTB16*, *KLRB1*, *IL18R1* and *IL23R* in CD4 T cells from non-inflamed and inflamed tissue samples compared to healthy controls (Supplemental Fig. 7B), supporting the idea that T$_{MIC}$ cells are present in inflamed colitic tissue. Further, compared to cells isolated from non-inflamed human tissue, T$_{MIC}$ cells in UC showed higher expression of inhibitory receptors like TIGIT, TIM3, LAG3, and CD39 (Fig. 5B), even more pronounced than their CD161$^-$ and CD161$^{int}$ counterparts (Supplemental Fig. 7C). These observations suggest that human T$_{MIC}$ cells are triggered in the context of IBD, which was also supported by increased expression of CTLA4 and TIGIT in CD4 T cells from inflamed IBD tissue when comparing cells expressing the aforementioned T$_{MIC}$-associated gene module (Supplemental Fig. 7D).

Taken together with the high effector potential and microbe-responsiveness displayed by T$_{MIC}$ cells, these findings suggest that T$_{MIC}$ cells might be involved in UC pathology.

## T$_{MIC}$ cells contribute to colon pathology in murine models of colitis
To determine whether microbe-reactive T cells can contribute to colonic pathology, we returned to the Cbir1 transgenic mouse model, which recognizes a microbe present in the normal mouse microbiota. While transferring isolated tissue-resident cells into another host to study their behavior would be the ideal experiment, as with other tissue-resident cell populations, DN cells isolated from the colon would not repopulate the gut of a recipient mouse, even in the case of Rag−/− where there is no competition from endogenous T cells (Supplemental Fig. 8A–C). To get around this issue and study the innate-like T cell population in isolation, we compared Cbir1 Rag−/− mice treated with anti-CD4 to deplete Th and lymphoid tissue inducer cells (~90% DN T cells) with anti-CD4 treated Rag−/− (no T cells) mice (Supplemental Fig. 8D, E). To understand whether these cells could contribute to pathogenesis when they encounter their antigen in a proinflammatory environment, we used the dextran sulfate sodium (DSS) model of colitis, which results in barrier breakdown and bacterial translocation. When challenged with DSS, CD4−depleted Cbir1 mice lost significantly more weight than their CD4−depleted Rag−/− littermates at the peak of the disease (Fig. 5C and D). Hallmarks of systemic disease were associated with a trend toward increased colon pathology marked by epithelial damage and crypt abscesses (Fig. 5F, G). Systemic inflammation was further marked by increased splenomegaly in the Cbir Rag−/− mice, bearing T$_{MIC}$ (Fig. 5E). To understand the differences

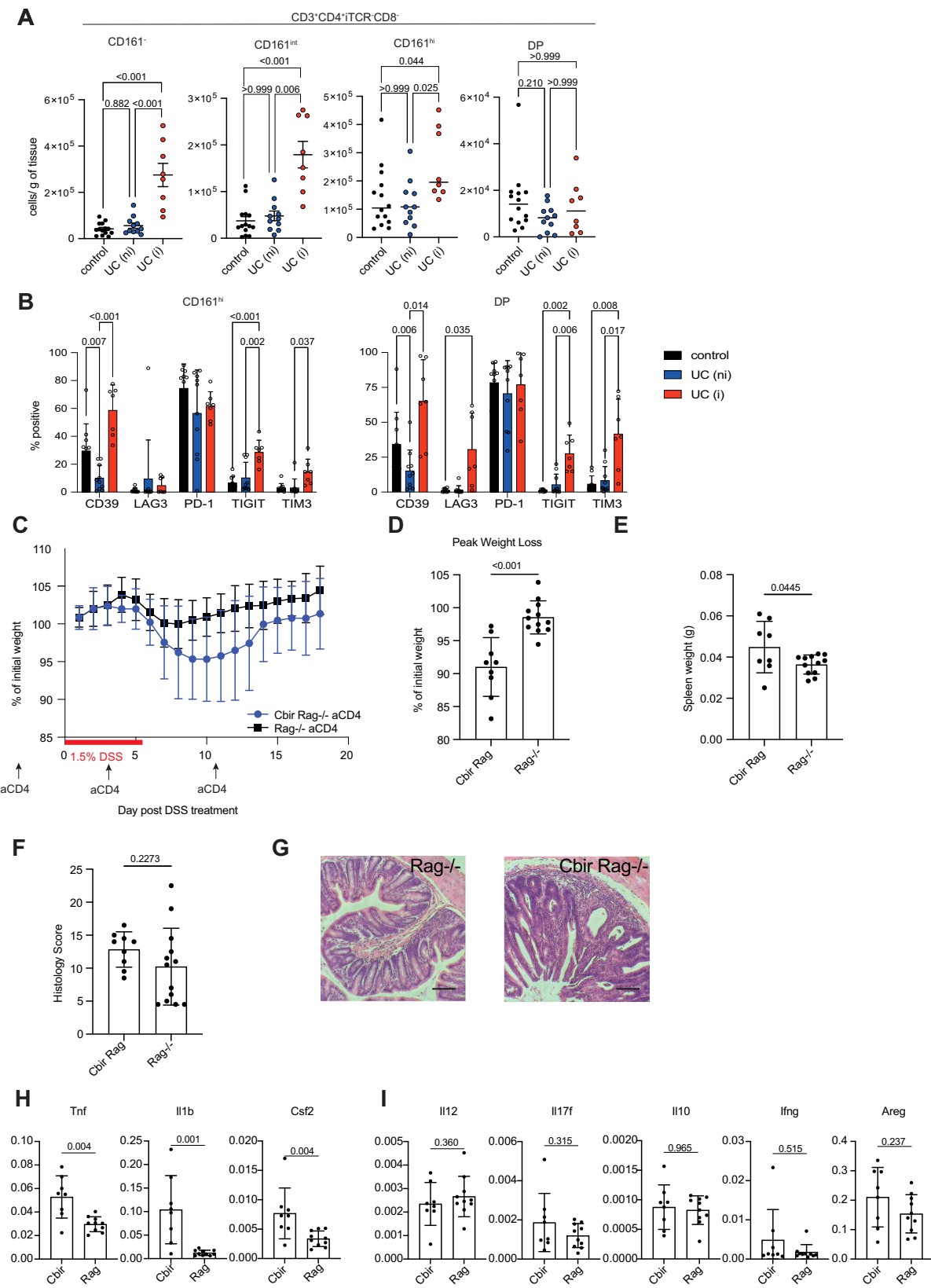

in disease on a tissue level, RNA was isolated from colonic tissue, and qPCR was performed for genes of interest associated with T_MIC. The presence of T_MIC in the Cbir mice resulted in significantly increased levels of *Tnf, Il1b, and Csf2* (GM-CSF) (Fig. 5H) but not other pro-inflammatory cytokines or the repair-associated gene *Areg* (Fig. 5I)

suggesting a specific response module in the context of DSS. Taken together, these data suggest that we have identified an as-yet unappreciated population of commensal and cytokine-responsive T cells that can contribute to intestinal pathogenesis in the context of barrier breakdown.

**Fig. 5 | T<sub>MIC</sub> cells are present in the inflamed tissue of human UC patients and exacerbate pathology in a murine colitis model.** See also Supplemental Fig.s 7 and 8. **A** LPMCs were isolated from resections of normal tissue from colorectal cancer patients (control, n = 10 biologically independent samples) or from biopsies obtained from UC patients. UC samples were classified as "non-inflamed" (UC(ni), n = 10 biologically independent samples) or inflamed (UC(i), n = 8 biologically independent samples) based on their UCEIS score. All samples were weighed, the total numbers of iTCR (Vα7.2, Vα24-Jα18, TCRγδ)-negative CD4 T cells expressing CD161⁻, CD161ⁱⁿᵗ, CD161ʰⁱ and DP (double-positive for CD161 and CD56) CD4 T cells in each sample recorded by flow cytometry and then normalized to the weight of the respective sample in grams. **B** Bar plots summarizing the expression of CD39, LAG3, PD-1, TIGIT, and TIM3 as determined by flow cytometry on the human T<sub>MIC</sub> cells from control (n = 10 biologically independent samples), UC (ni) (n = 10 biologically independent samples) or UC (i) (n = 7 biologically independent samples)

samples. **A**, **B** Data points were pooled from independent experiments using one or two human samples each. **C** Weight loss curve of Cbir Rag−/− or Rag−/− littermates pre-treated with anti-CD4 antibody and given 1.5% dextran sulfate sodium DSS in drinking water for 6 days. **D** Quantification of peak weight loss for each animal in the Cbir Rag−/− and Rag−/− controls. **E** Spleen weights at DSS experiment endpoint. **F** Quantification of colon histology score at DSS experiment endpoint. Each point represents one individual mouse. n = 9 Cbir, n = 12 Rag−/−, showing a combination of two independent experiments. **G** Example H&E staining showing unique features of epithelial damage in Cbir Rag−/− after DSS. Scale bar = 50 μm representative of n = 9 Cbir, n = 12 Rag−/− from two independent experiments. **H**, **I** qPCR of gene expression relative to *Hprt* for colonic tissue isolated on day 18 after DSS administration. Each point represents an individual mouse. n = 9 Cbir, n = 12 Rag−/−, showing a combination of two independent experiments. Source data are provided as a Source Data file.

## Discussion

Commensal-immune interactions are key in maintaining homeostasis in the intestine. Using human tissue samples and mouse models, our data demonstrate the acquisition of innate-like features by commensal-reactive, MHC-II-restricted T cells in the colon. These characteristics place human and murine T<sub>MIC</sub> cells on the expanding array of T cell populations bridging innate and adaptive immunity. Similar to T cell subsets described in itk−/− mice and fetal human tissues[23,24], T<sub>MIC</sub> lie between established innate-like T cell subsets such as MAIT and iNKT cells and classic tissue-resident memory T cells as a population expressing innate-like factors such as PLZF and high levels of CD161 (in humans), while featuring a diverse, MHC-II restricted TCR repertoire at the same time. Located at mucosal barriers, T<sub>MIC</sub> cells are likely to play an important role in microbial surveillance and shaping the local cytokine milieu.

T<sub>MIC</sub> cells are potent producers of effector molecules including several key Th17 cytokines. This is in line with previous studies that described the broader population of human colonic CD161 + CD4 T cells as a Th17 population in the context of Crohn's disease[25] and observed IL-17 production by murine DN T cells in response to intracellular pathogens[50] and human DN T cells in lupus[52]. Our findings expand on this, as we demonstrate that especially the CD161ʰⁱ subset is not restricted to Th17 functions but displays a dual Th1/Th17 effector profile much like human innate-like T cells[56] and is capable of producing factors associated with cytotoxicity. Notably, despite the differences in CD4 expression, these functional properties were conserved in murine T<sub>MIC</sub> cells. As we were analyzing cells derived from normal, uninflamed tissue in this study, it is conceivable, that in the context of human inflammatory disease, T<sub>MIC</sub> cells would get skewed more towards a Th17-phenotype while our findings represent a more homeostatic state representative of healthy tissue. Indeed, several recent publications have found that in human intestinal CD4 T cells, effector phenotypes exist as a gradient across the population rather than as distinct subsets[57–59], supporting the idea that many of these cells have the potential to produce a wide range of effector molecules.

Mucosal tissues are rich in T<sub>RM</sub> cells which arise in response to infection or vaccination and confer local protection in the event of re-encounter with a given pathogen. Interestingly, analyses of single cell sequencing data revealed that the expression of an innate-like gene expression profile does not correlate tightly with the expression of established T<sub>RM</sub> gene signatures, indicating that the T<sub>MIC</sub> phenotype constitutes a special status and not part of the general T<sub>RM</sub> program in the colon. While the pathways leading to T<sub>MIC</sub> formation are not fully understood at the moment, it is worth pointing out that most T<sub>RM</sub> models make use of acute infections such as LCMV and HSV or precisely controlled challenges with model antigens or vaccines to induce T<sub>RM</sub> cells[46,47,60,61], meaning that antigen exposure is intense and transient. In contrast, T cells responding to commensal microbes would be triggered with limited amounts of antigen but multiple times over a prolonged period of time or even constitutively. While future studies

are required to fully address this question, this model would be supported by the observed differences between microbe-reactive Cbir- or Hh-TCR transgenic T cells. While the former respond to a ubiquitous commensal antigen and acquire a T<sub>MIC</sub> phenotype in the gut, the latter, responding to a bacterium not normally present, display a typical CD4 phenotype at steady-state and maintains a conventional CD4 phenotype when transferred into an infected host[11]. Further work with other microbe-reactive TCR transgenics[9] may shed light on the ability of other microbes to drive this phenotype.

Existing data indicate that the local factors specific to mucosal tissues play an essential role in supporting the T<sub>MIC</sub> phenotype as Cbir T cells located in non-mucosal organs do not have an innate-like phenotype. While a few T<sub>MIC</sub> cells can be identified in the thymus of B6 and Cbir mice, the accumulation of cells with this phenotype over the life course is likely a result of adaption to local conditions. We cannot disregard the possibility that some cells acquire this phenotype in the tissue, and this may be supported by the finding that murine T<sub>MIC</sub> cells can be found in all mucosal tissues examined. While future experiments are required to identify which antigen-presenting cells, cytokines and signaling mechanisms are required for T<sub>MIC</sub> maintenance, we hypothesize that local myeloid populations in the lamina propria could be involved. These antigen-presenting cells have the potential to stimulate T<sub>MIC</sub> cells through TCR and IL-23R simultaneously while cytokines from epithelial cells such as IL-18 may tune T<sub>MIC</sub> cells to produce a different set of responses.

Our results confirmed several other studies reporting the colitogenic potential of microbe-reactive T cells[62–64] and are further supported by the fact that human T<sub>MIC</sub> cells are present in inflamed tissue of ulcerative colitis patients in normal or even enhanced numbers. However, the role T<sub>MIC</sub> cells play in the lamina propria likely goes beyond that. Innate-like T cells including MAIT cells were recently shown to possess tissue repair capacities[39–41] in humans and mice and since T<sub>MIC</sub> cells share transcriptional features with these populations, T<sub>MIC</sub> cells could perform similar function under the right circumstances. In MAITs, whether activation is induced by TCR or cytokine-signaling is critical in determining if they express repair-associated factors or a pure pro-inflammatory program, respectively[38,39]. T<sub>MIC</sub> cells might operate in a similar way and, in a steady state, triggered in a limited fashion by commensal-dependent TCR-mediated signals, could actually contribute to tissue homeostasis and the repair of limited injuries. In contrast, in the context of wide-spread inflammation, e.g., after a major breach of the epithelial barrier like in the DSS colitis model we tested, cytokine- and combined cytokine- and TCR-mediated activation would likely dominate, leading to massive production of Th1 and Th17 effector molecules contributing to colitis.

T cells with a T<sub>MIC</sub> phenotype are not a rare cell population in the lamina propria and are especially abundant in humans. Given their high effector and colitogenic potential as well as their responsiveness to microbes, future studies addressing how exactly these cells are induced and regulated and how they

behave in the steady state are needed. Fully understanding their function could lead to new therapeutic targets in relevant human diseases like IBD and checkpoint-induced colitis and could have important implications for patients undergoing any kind of microbiome therapy or suffering from conditions associated with alterations in the microbiome.

## Methods

### Ethical approval

All experiments in this study have been conducted in accordance with international and local ethical standards. Human tissue samples were collected with appropriate patient consent and NHS REC provided ethical approval (reference number 16/YH/0247). Human donors did not receive compensation. Experiments involving mice were conducted in accordance with local animal care committees (UK Scientific Procedures Act of 1986). The project license (P508FFA1F) governing the mouse studies was reviewed by the University of Oxford's Animal Welfare and Ethical Review Board and approved by the Home Office of his Majesty's Government.

### Experimental model and subject details

**Human samples.** Normal adjacent tissue from colorectal cancer (CRC) patients who were undergoing surgery was collected by the TGU biobank. Biopsies were obtained from ulcerative colitis (UC) patients or healthy controls attending the John Radcliffe hospital. All tissue samples were collected with appropriate patient consent and NHS REC provided ethical approval (reference number 16/YH/0247).

Characteristics of the CRC patients

| CRC patients | $n = 49$ |
|---|---|
| Age (years, average, SD) | $68 \pm 12$ |
| Sex (Male/Female) | 28/21 |
| Time since diagnosis (months, average, SD) | $1 \pm 4$ |

Characteristics of the healthy controls

| Healthy controls | $n = 10$ |
|---|---|
| Age (years, average, SD) | $51 \pm 9$ |
| Sex (Male/Female) | 7/3 |

Characteristics of the UC patients

| UC patients | | $n = 20$ |
|---|---|---|
| Inflamed | | $n = 9$ |
| | Age (Average, SD) | $38 \pm 19$ |
| | Sex (Male/Female) | 4/5 |
| | Time since diagnosis (years, average, SD) | $14 \pm 12$ |
| | UCEIS score (average, range) | 3.7, [2–6] |
| Uninflamed | | $n = 11$ |
| | Age (Average, SD) | $53 \pm 11$ |
| | Sex (Male/Female) | 4/7 |
| | Time since diagnosis (years, average, SD) | $17 \pm 13$ |
| | UCEIS score (average, range) | 0.1, [0–1] |

**Mice.** Mice were bred and maintained in the University of Oxford specific pathogen-free (SPF) animal facilities. Experiments were conducted in accordance with local animal care committees (UK Scientific Procedures Act of 1986) under the project license P508FFA1F. Mice were routinely screened for the absence of pathogens and were kept in individually ventilated cages with environmental enrichment at 20–24 °C, 45–65% humidity with a 12 h light/dark cycle (7am–7pm)

with half an hour dawn and dusk period. Specific pathogen-free mice were fed SDS RM3 diet (LBS Biotechnology) *ad libitum*, and germ-free mice were fed the same RM3 diet irradiated to 50kGy. The C57BL/6 age cohort was purchased from Charles River. B2Mko mice were obtained from The Jackson Laboratory and maintained by Oliver Harrison at the Benaroya Research Institute. Cbir1 TCR transgenic mice were a kind gift from Charles Elson III. *Il23r*gfp/+ were obtained from Daniel Cua (Merck Research Laboratories, Palo Alto, USA). Hh7-5 TCR transgenic mice were a kind gift from Dan Littman. Backcrosses to B6J were verified by SNP analysis performed by Transnetyx. Mice were age and sex-matched, using equal numbers of each sex where possible with the exception of DSS experiments. DSS experiments were performed with females to reduce cage numbers and potential confounding pathology from fighting. Mice were divided into cages, ensuring all genotypes were present in each cage, and experimenters were blinded to genotypes for the duration of the experiment.

### Method details

**Processing of CRC tissue resections.** Fat and muscle tissue were superficially removed using sterile scissors and forceps before the mucosa was washed in 1 mM DTT (Sigma-Aldrich) dissolved in PBS (GIBCO) supplemented with 40 µg/ml genticin (Thermo Fisher), 10 µg/ml ciprofloxacin, 0.025 µg/ml amphotericin B and 100 U/ml penicillin, 0.1 mg/ml streptomycin (all from Sigma-Aldrich) for 15 min in a shaker set to 200 rpm at 37 °C. To remove epithelial cells, samples were subsequently washed three times in the same PBS-based buffer containing 5 mM EDTA (VWR). Finally, specimens were weighted, cut into small pieces, and either directly digested or stored on 1 ml freezing medium (90% FCS, 10% DMSO) at −80° for later usage (0.2–0.4 g per vial).

### Direct digestion of resection-derived tissue

For the assays aiming to detect microbe-reactive CD4 T cells, resection-derived tissues were directly digested by transferring the dissected tissue into an RPMI 1640-based digestion medium containing 10% FCS (Sigma-Aldrich), 40 µg/ml genticin (Thermo Fisher), 10 µg/ml ciprofloxacin, 0.025 µg/ml amphotericin B, 100 U/ml penicillin, 0.1 mg/ml streptomycin (all from Sigma-Aldrich), 0.1 mg/ml collagenase A and 0.01 mg/ml DNase I (both from Roche). HEPES (10 mM, GIBCO) was freshly added to the buffer before the digestion. The specimens were incubated for 20 min in a shaker set to 200 rpm at 37° Celsius and filtered over a 100 µm Cell strainer to isolate liberated cells. This process was then repeated several times until the tissue had been fully digested and no more additional cells could be obtained. Liberated cells were collected and pooled in R10 medium (RPMI 1640 containing 10% FCS, 100 U/ml penicillin, 0.1 mg/ml streptomycin, and 2 mM L-glutamine) and stored at 4 °C. To enrich mononuclear cells, digests were centrifuged for 20 min (400 g, room temperature, no brake) in a two-layer Percoll (Sigma-Aldrich) gradient and collected at the 40%/80% interphase.

### Generation of single-cell suspensions from stored tissue and biopsies

To remove the DMSO-containing freezing medium, resection-derived tissue samples were washed over a 70 µm Cell strainer with pre-warmed R10 medium. Washed resection specimens or biopsies obtained from UC patients or healthy controls were transferred into gentleMACS C tubes (Miltenyi) and 5 ml of an RPMI 1640-based digestion medium containing 1 mg/ml collagenase D and 0.01 mg/ml DNase I (both from Roche) was added. Samples were homogenized on the gentleMACS (Miltenyi) with the brain01_02 program and incubated for 60 min in a shaker set to 200 rpm at 37 °C. All specimens were then fully disrupted on the gentleMACS by running program B01 and the content of the C tubes was filtered into R10 medium over 70 µm cell strainers. Cell was washed two times in R10 medium (10 min, 300×*g*,

4°C) and again, mononuclear cells were enriched by density Percoll (Sigma-Aldrich) centrifugation (20 min, 400 × *g*, room temperature, no brake) and collected at the 40%/80% interphase.

**Detection of microbe-reactive human CD4 T cells.** Mononuclear cells from directly digested colonic resections were counted and adjusted to $10^7$ cells/ml in R10 medium and plated out on 96U plates, $10^6$ cells per well. In some experiments, cells were incubated with 10 µg/ml ULTRA-LEAF purified mouse antihuman MHC-II (Tü39), CD1d (51.1), MR1(26.5), IgG2αк (MOPC-173 for MHC-II and MR1) or IgG2bk (MCP11 for CD1d) control antibodies or a mixture of ULTRA-LEAF purified antihuman IL-12p40 (C8.9, 5 µg/ml) and anti IL-18 (W17071A, 5 µg/ml) or (Biolegend) for 30 min at 37°C to evaluate-dependency of the observed responses on these receptors and cytokines. Next, cells were mixed with heat-killed microbes: *E. coli* Nissle (Ardeypharm GmbH), *S. aureus* (National collection of type cultures 6571) and *C. albicans* (InvivoGen) in a 1:10 ratio. After two hours of incubation at 37°C, Brefeldin A (Invitrogen) was added to the cells to block cytokine release and then incubation as continued for 6 additional hours. Finally, cells were washed with PBS (2 min, 300 × *g*, room temperature), pelleted by centrifugation and microbe-reactive cells were detected by FACS staining for CD154 (24–31) and TNF (Mab11). Both antibodies were purchased from Biolegend and used at 1:50.

**TCR and cytokine stimulation of human CD4 T cells**
To provide TCR stimulation to human CD4 T cells, NUNC Maxisorp plates (BioLegend) were coated with purified antihuman CD3 antibodies (BioLegend) diluted to 1.25 µg/ml in sterile PBS for 2 h at 37°C or at 4°C overnight. Control wells were filled with PBS only. Plates were washed two times with PBS and one time with R10 medium before cells were added.

Mononuclear cells were adjusted to either $4 × 10^6$ cells/ml in R10 medium and 100 µl cell suspensions were added to the appropriate well of the coated plates. Purified antihuman CD28 antibodies (BioLegend) and added to all wells previously coated with anti CD3 antibodies (OKT3) to a final concentration of 1 µg/ml. Recombinant human IL-23 (Miltenyi) was diluted to 400 ng/ml in R10 medium and added to the cells to a final concentration of 50 ng/ml.

For stimulation with IL-12 (Miltenyi) and IL-18 (BioLegend), cells were plated out at a concentration of $10^7$ cells/ml in R10 medium on regular 96U plates and mixed with recombinant cytokines (final concentration 50 ng/ml for both IL-12 and IL-18). In all experiments, some cells were left untreated to determine baseline expression of the analyzed effector molecules.

Cells were incubated with different combinations of antibodies or cytokines for 24 h total. Brefeldin A solution (Invitrogen) was added to all well for the last 4 h to prevent cytokine release.

**FACS**
Single-cell suspensions were initially stained with the LIVE/DEAD Fixable Near IR Dead Cell dye (Invitrogen) diluted 1:1000 in PBS for 20 min at room temperature. Cells were then washed (2 min, 300×g, room temperature) and stained in PBS supplemented with 0.5% FCS and 2 mM EDTA with the antibodies purchased from Biolegend, BD, Miltenyi, Invitrogen, or Thermo Fisher for 20–30 min at room temperature.

Prior to subsequent staining steps or acquisition, cells were fixed in 2% Formaldehyde (Sigma-Aldrich) for 10 min at room temperature. For intracellular cytokine staining, cells were permeabilized and stained with the appropriate antibodies using the BD Cytofix/Cytoperm Kit following the manufacturer's instructions. Intranuclear staining for transcription factors was performed using the Foxp3/ Transcription Factor Staining Buffer Kit (Invitrogen) according to the manufacturer's instructions. The following antibodies were used to stain human cells:

FITC-CCR7 (clone: G043H7, Biolegend Cat # 353215, 1:100), BV570-CD3 (clone: UCHT1, Biolegend Cat # 300436, 1:100), PE/Dazzle 594-CD3 (clone: UCHT1, Biolegend Cat # 980006, 1:100), PerCp/Cy5.5-CD3 (clone: UCHT1, Biolegend Cat # 300430, 1:100), BV605- CD4 (clone: OKT4, Biolegend Cat # 317438, 1:100), BV650-CD4 (clone: OKT4, Biolegend Cat # 317436, 1:100), BV650-CD8 (clone: SK1, Biolegend Cat # 344730, 1:100), BV421-CD39 (clone: A1, Biolegend Cat # 328214, 1:200), PE/Dazzle594-CD45RA (clone: HI100, Biolegend Cat # 304145, 1:100), PE/Cy7-CD45RO (clone: UCHL1, Biolegend Cat # 304229, 1:50), BV421-CD56 (clone: HCD56, Biolegend Cat # 318328, 1:100), BV480-CD56 (clone: NCAM16.2, BD Bioscienes Cat # 566124, 1:100), PE/Cy7-CD154 (clone: 24–31, Biolegend Cat # 310832, 1:50), BV421-CD161 (clone: HP-3G10, Biolegend Cat # 339914, 1:100), PE-CD161 (clone: 191B8, Miltenyi Cat # 130-113-593, 1:200), APC-GzmB (clone: GB11, Invitrogen, 1:100), Alex Fluor 700-GzmB (clone: QA16A02, Biolegend, 1:100), PE/Vio770-ICOS (clone: REA192, Milteny, 1:50), Alex Fluor 700-IFNγ (clone: 4 S.B3, Biolegend, 1:100), BV711-IFNγ (clone: 4 S.B3, Biolegend, 1:100), BV785-IFNγ (clone: 4 S.B3, Biolegend, 1:100), FITC-IFNγ (clone: 45-15, Miltenyi, 1:100), Alexa Fluor 647 mouse IgG1κ (clone: MOPC-21, BD Biosciences Cat # 557732, 1:50), PE mouse IgG2bκ (clone: 27–35, BD Biosciences Cat # 555058, 1:50), BV421-IL-17A (clone: BL168) Biolegend Cat # 512322, 1:100), FITC-IL-17A (clone: BL168, Biolegend Cat # 512304, 1:00), PE-IL17F (clone: eBio18F10-Invitrogen Cat # 12-7471-82, 1:100), PE/Cy7-IL17F (clone: SHLR17, Thermo Fisher Cat # 12-7169-42, 1:100), PE/Cy7-IL18Rα (clone: H44, Biolegend Cat # 31381, 1:125), APC-IL18Rα (clone: H44, Biolegend Cat # 313814, 1:100), PerCp/eFluor 710-IL-22 (clone: 22URTI, Thermo Fisher Cat # 46-7229-42, 1:50), PE/Cy7-LAG3 (clone: 7H2C65, Biolegend Cat # 369208, 1:100), BV785-PD-1 (clone: 29 F.1A12, Biolegend Cat # 135225, 1:50), Alex Fluor 647-PLZF (clone: R17-809, BD Biosciences Cat # 563490, 1:50), BV711-TCR Vα7.2 (clone: 3C10, Biolegend Cat # 351732, 1:100), FITC-TCR Vα7.2 (clone: 3C10, Biolegend Cat # 351704, 1:50), PerCp/Cy5.5-TCR Vα7.2 (clone: 3C10, Biolegend Cat # 351710, 1:100), BV711-TCR Vα24-Jα18 (clone: 6B11, Biolegend Cat # 342922, 1:100), PerCp/Cy5.5-TCR Vα24-Jα18 (clone: 6B11, Biolegend Cat # 342914, 1:100), PerCp/Cy5.5-TCRγδ (clone: B1, Biolegend Cat # 331224, 1:100), APC/Fire 750-TCRγδ (clone: B1, Biolegend Cat # 331228, 1:100), APC-TIGIT (clone: A15153G, Biolegend Cat # 372706, 1:100), BV605-TIM3 (clone: F38-2E2, Biolegend Cat # 345018, 1:50),PerCp/Cy5.5-TNF (clone: MAb11, Biolegend Cat # 502926, 1:50), PE-RoRγt (clone: Q21-559, BD Biosciences Cat # 563081, 1:50).

The following antibodies were used to stain murine cells:
PerCP/Cyanine5.5-CD45R/B220 (clone: RA3-6B2, Biolegend Cat # 103236, 1:200), PerCP/Cyanine5.5-CD11b (clone: M1/70, Biolegend # 101228, 1:200), PerCP/Cyanine5.5-CD11c Antibody (clone: N418, Biolegend Cat # 117328, 1:200), APC-TCR γ/δ Antibody (clone: GL3, Biolegend Cat # 118116, 1:200), Alexa Fluor® 700-CD45 (clone: 30-F11, Biolegend Cat # 103128, 1:300), Brilliant Violet 421™-CD127 (IL-7Rα) (clone: A7R34, Biolegend Cat # 135023, 1:100), Brilliant Violet 605™-CD8a (clone: 53-6.7, Biolegend Cat # 100743, 1:300), Brilliant Violet 785™-CD4 (clone: RM4-5, Biolegend Cat # 100552, 1:200), FITC-TCR β chain (clone: H57-597, Biolegend Cat # 109206, 1:200), Alexa Fluor 488-PLZF (clone: Mags.21F7, Life Technologies Cat # 53-9320-82, 1:100), PE/Dazzle™ 594-CD8a (clone: 53-6.7, Biolegend Cat # 100762, 1:300), PE/Cyanine7-TCR β (clone: H57-597, Biolegend Cat # 109222, 1:200), APC-CD127 (IL-7Rα) (clone: A7R34, Biolegend Cat # 135012, 1:100), BV421-RORγt (clone: Q31-378, BD Biosciences Cat # 562894, 1:200), Brilliant Violet 605™-CD4 (clone: RM4-5, Biolegend Cat # 100548, 1:200), Brilliant Violet 605™-CD8a (clone: 53-6.7, Biolegend Cat # 100744, 1:300), Brilliant Violet 650™-CD25 (clone: PC61, Biolegend Cat # 102037, 1:200), Brilliant Violet 785™-CD45 Antibody (clone: 30-F11, Biolegend Cat # 103149, 1:300), Brilliant Violet 650™-NK1.1 (clone: PK136, Biolegend Cat # 108735, 1:200), Alexa Fluor® 700-TCR β (clone: H57-597, Biolegend Cat # 109224, 1:200), PE/Cyanine7-TCR γ/δ (clone: GL3, Biolegend Cat # 118124, 1:200), Brilliant Violet 421-CD8a

(clone: 53-6.7, Biolegend Cat # 100737, 1:300), Brilliant Violet 605-CD11c (clone: N418) Biolegend Cat # 117334, 1:200), PE-CD218a (IL-18Ra) (clone: P3TUNYA, Life Technologies Cat # 12-5183-80, 1:100) PE- or APC-labeled mouse CD1d- and MR1- tetramers were obtained from the NIH Tetramer Core Facility and titrated per lot for binding to spleen and liver NKTs and MAITs, generally 1:1000 for CD1d and 1:500 for MR1.

All cells were acquired on a LSR II or Fortessa (BD) using FACS Diva Software (BD); data were analyzed using FlowJo software (Treestar).

**Nucleic acid extraction and bulk TCR sequencing of human cells**
TRIzol (Thermo Fisher) nucleic acid extraction was used to extract high-purity RNA from sorted human colonic CD4 T cells. Briefly, after sorting, cells were centrifuged (500×*g*, 5 min), resuspended in 1 ml TRIzol, then frozen at −80 °C until RNA extraction. For RNA extraction samples were brought to room temperature, mixed with 200 μl chloroform (Sigma-Aldrich) and centrifuged at 12,500 rpm for 5 min. 500 μl of the aqueous phase was taken and RNA extracted using the RNAdvance Tissue Isolation kit (Agencourt). RNA was assessed for concentration and purity using the RNA Pico assay on a 2100 Bioanalyzer instrument (both Agilent). Bulk TCR repertoire sequencing was performed using the amplicon-rescued multiplex (ARM)-PCR method (iRepertoire Inc). Library generation was performed in-house according to the manufacturer's instructions. In brief, the extracted RNA, enzyme mix, and barcoded primer reaction mix were mixed on ice, followed by a combined RT-PCR and initial amplification step in the thermal cycler using the iRepertoire low-input protocol. Next, the products were purified using the kit's solid phase reverse immobilization (SPRI) beads and ethanol washes, then eluted in water. This product was then combined with enzyme mix and universal primers for the second amplification step. The final product was purified again using SPRI beads and eluted in water. The quality, size distribution, concentration, and presence of contaminating primer dimers of the final product was assessed using several QC steps, including identification of a clear band of appropriate size on agarose gel electrophoresis, a spectral photometer (Nanodrop, Thermofisher Scientific), and the DNA 1000 kit using a 2100 Bioanalyzer instrument (Agilent). Libraries were quantified using the KAPA Library Quantification Kit (Roche) on a CFX96 Thermal Cycler instrument (Bio-Rad) before equimolar pooling. Samples were submitted to the Oxford Genomics Centre where a PhiX library spike-in was added (10%) due to the low diversity of the TCR library, before 300 bp paired-end sequencing on an Illumina MiSeq instrument (WTCHG,University of Oxford) was run.

**Bulk TCR repertoire analysis.** Data processing of TCR repertoire libraries was performed using the iRepertoire analysis pipeline. In brief, reads were demultiplexed based on the 6-N molecular barcode associated with the sample. Low-quality reads were trimmed (removing anything with a Phred score of less than 30), and R1 and R2 reads were overlapped and stitched. Only stitched reads where identity within the overlapped portions was 100% were included in downstream analysis. Reads were then mapped to the IMGT database, and only reads that map to reference sequences and contain canonical CDR3 motifs were included for further analysis. Finally, several filters (see irepertoire.com/irweb-technical-notes) were applied to remove sequencing artefacts, PCR artefacts, insertion, deletion, substitution errors, and low frequency (*n* = 1) reads. Initial data analysis was performed using the iRweb data analysis platform (iRepertoire, Inc., USA). Additional analysis and generation of plots was performed using SeeTCR (friedmanlab.weizmann.ac.il/SeeTCR).

**BD Rhapsody targeted single-cell transcriptomics.** Lamina propria mononuclear cells were isolated from three donors, stimulated with IL-12/18, plate-bound αCD3 or a combination of both as described above overnight. Samples were stained in parallel with oligonucleotide-

conjugated Sample Tags from the BD Human Single-Cell Multiplexing Kit and a panel of 50 Abseq antibodies in BD stain buffer following the manufacturer's protocol.

The following BD AbSeq antibodies were purchased from BD Biosciences and used at 2 μl per sample: CD4 (clone: RPA-T4), CD7 (clone: M-T701), CD9 (clone: M-L13), CD25 (clone: M-A251), CD26 (clone: M-A261), CD27 (clone: M-T271), CD39 (clone: TU66), CD45RA (clone: HI100), CD45RO (clone: UCHL1), CD49a (clone: SR84), CD49d (clone: 9F10), CD56 (clone: NCAM16.2), CD58 (clone: 1C3), CD62L (clone: DREG-56), CD69 (clone: FN50), CD72 (clone: J4-117), CD73 (clone: AD2), CD83 (clone: HB15e), CD94 (clone: HP-3D9), CD95 (clone: DX2), CD103 (clone: Ber-ACT8), CD119 (clone: GIR-208), CD122 (clone: MIK-BETA3), CD123 (clone: 7G3), CD124 (clone: hIL4R-M57), CD126 (clone: M5), CD127 (clone: HIL-7R-M21), CD131 (clone: 3D7), CD132 (clone: TUGh4), CD134 (clone: ACT35), CD137 (clone: 4B4-1), CD140A (clone: αR1), CD140B (clone: 28D4), CD154 (clone: TRAP1), CD178 (clone: NOK-1), CD181 (clone: 5A12), CD183 (clone: 1C6/CXCR3), CD192 (clone: LS132.1D9), CD197(clone: 3D12), CD212 (clone: 2.4E6), CD215 (clone: JM7A4), CD278 (clone: DX29), CD294 (clone: BM16), CD335 (clone: 9E2/Nkp46), GITR (clone: V27-580), IL-21R (clone: 17A12), Itgb7 (clone: FIB504), LAG3 (clone: T47-530).

In addition the following customized AbSeq antibodies were provided by BD Biosciences and used at 2 μl/test: CD161 (clone: 191B8, based on Miltenyi: RRID:AB_871628) and CD196 (clone: G034E3 based on Biolegend: RRID:AB_10918625).

Subsequently, cells were stained with a smaller panel of fluorescently labeled sorting antibodies (CD4, CD8, Vα7.2, TCRγδ, Vα24-Jα18, CD45) and LIVE/DEAD Fixable Near IR Dead Cell dye as for FACS experiments. 20,000– 40,000 cells per donor and stimulatory condition were sorted on a BD ARIA III as live, CD45+ CD4+ CD8− Vα7.2− Vα24-Jα18− TCRγδ− in the Experimental Medicine Division Flow Cytometry Facility. Sorted cells were spun down (300×*g*, 5 min) and resuspended in 200 μl chilled BD sample buffer, and 20,000 cells were pooled together from all 12 samples and subsequently loaded onto a BD Rhapsody cartridge. Single cell capture and cDNA synthesis with a BD Rhapsody express system were performed using the manufacturer's reagents and protocols. In brief, the process included cell-capture with beads in microwell plate, followed by cell lysis, bead recovery, cDNA synthesis and library preparation using using the BD Rhapsody Targeted mRNA and Abseq Amplification kit.

Separate libraries were prepared for sample tags, Abseq, and targeted mRNA using a basic immune panel (BD Rhapsody Immune response Panel Hs) in combination with custom panel covering 145 additional genes (Supplementary Data 6). Importantly, the latter included genes associated with tissue-repair in a population of unconventional T cells in the murine skin[41] which were also be shown to be expressed in murine and human MAIT cells[38,39]. Sequencing of the pooled libraries was completed on a NovaSeq6000 (Illumina, San Diego, CA) at Novogene (Cambridge, UK).

**BD Rhapsody data analysis.** The FASTQ-files generated from the Rhapsody experiment were uploaded onto the Seven Bridges Genomics online platform together with FASTA-files containing the sequence information about the mRNA and Abseq targets and were subjected to the BD Rhapsody Targeted analysis pipeline. Data analysis was performed in SeqGeq (BD). Briefly, quality control was performed by gating out events with low gene expression, samples were demultiplexed using the Lex-BDSMK plugin to separate cells from the different stimulation conditions and CD161−, CD161int, and CD161hi cells were identified by gating using DNA-barcoded antibodies for CD161 and CD56. Gene expression profiles for the CD161hi cells were exported and z-scores were calculated for tissue-repair-associated genes upon cytokine, TCR, or combined stimulation. Expression of the whole tissue-repair gene signature was analyzed by gene set enrichment analysis using the fgsea R package (see below).

## Gene signature expression analysis in published datasets

Previously published colonic single-cell data[48,49] were re-analyzed using version 4 of the Seurat R package[65] following the package's vignette. In brief, cells with abnormally high features numbers or high percentages of mitochondrial genes were removed, data were normalized (*method = LogNormalize* of the *NormalizeData* function) and scaled (ScaleData). Cell clusters were identified using the *FindClusters* and *RunUMAP* functions. To restrict the datasets to CD4 T cells, markers distinguishing the clusters were identified and visualized by the *FindAllMarkers* and *Featureplot* functions, and based on the expression of CD3E, CD4, CD8B, CD8A and ZBTB7, the datasets were subsetted. The Smillie dataset was further subsetted to retain only cells annotated as "Healthy", excluding cells isolated from IBD patients.

The processed gene counts were extracted (*GetAssayData* function) and an expression-based ranking was built for all genes in each cell by using the *AUCell_buildrankings* function from the AUCell R package[66]. Files containing lists of genes associated with $T_{RM}$ cells[46,47] or CD161-expressing cells[29] were loaded and processed to be compatible with AUCell in R using the *getGMT* and *setGeneSetNames* functions from the GSEABase R package[67]. An additional control dataset (150 Genes random) was generated by randomly selecting 150 genes from the respective datasets. Areas under the curve (AU) values were calculated using AUCells *AUCell_calcAUC* function. To analyze the relationship between the different gene sets, the AUCell values were exported and the Pearson correlation was calculated in Prism.

## Merging of human and murine RNAseq datasets

The merging of sequencing data from different sources[54] and different species[38,39] has been described before. In brief, a selection of RNAseq datasets from ImmGen[55] were merged with the human and murine RNA-sequencing datasets described in this study. In each dataset separately, zero-count and lowly expressed genes were initially removed from the raw read counts using the edgeR R package[68]. Raw counts were then log2-transformed using the *voom* function from the limma R package[69]. The murine orthologues of the genes present in the human dataset were identified using the *getLDS* function of the biomaRt package[70] and all datasets were merged based on common gene symbols. The *ComBat* function of the sva R package[71] was used to remove batch effects based on empirical Bayes methods[72]. The genes from the resulting merged dataset were filtered by variance (IQR > 0.75) and subjected to a principal component analysis and a hierarchical clustering analysis using the Euclidean distance metric.

## PCA-loading and GO-term analyses of the merged dataset

The *biplot* function of the PCAtools R package[73] was used to generate PCA plots based on the most variable genes of the previously merged dataset. The *plotloadings* function from the same package was then used to obtain lists of the genes driving the clustering of the cell populations along the major principal components. Lists of genes positively associated with either principal component 1 or 2 were then subjected to a GO-term analysis (the *ont* argument was set to "BP", *p* value cutoff 0.01, *q* value cutoff 0.05) to using the *enrichGO* function provided by the clusterprofiler R package[74] to predict biological processes associated with human CD161[hi] CD4 and murine DN T cells.

## Gene set enrichment analyses of the murine and human RNAseq datasets

Raw gene count data from human and murine RNA-sequencing experiments were loaded into R and genes not or only lowly (less than 10 total counts or not expressed in all samples from at least one experimental group) expressed were removed using basic R commands and the *filterByExpr* function of the edgeR R package[68]. Filtered gene counts were then processed, transformed, and normalized using the *DESeqDataSetFromMatrix*, *vst* and *DESeq* functions from the DESeq2 R package[75], respectively. The *results* function from the same

package was used to generate lists of genes differentially expressed between human CD161hi and CD161int/- CD4 T cells or murine wt or Cbir DN and CD4 T cells respectively. To conduct GSEA, a list of biological processes was obtained using the msigdbr R package[76] and the *fgseaMultilevel* function of the fgsea R package[77] was used to perform GSEA and calculate p-values, BH-adjusted p-values and normalized enrichment scores (NES). The *plotEnrichment* function from the same package was used to visualize the enrichment curves of relevant processes.

## Isolation of tissue leukocytes from mouse

**Colon.** Colon tissue was cut into ~1 cm pieces and incubated (2×) in RPMI containing 1% BSA (Sigma-Aldrich) and 5 mM EDTA (Sigma-Aldrich) in a 37 C shaking incubator. Remaining tissue was incubated in RPMI containing 1% BSA, 15 mM HEPES and 300 U/ml of Collagenase VIII (Sigma-Aldrich, St Louis, MO) to digest the remaining tissue. Cell populations were purified by 37.5% Percoll (GE Healthcare, Little Chalfont, UK) gradient centrifugation (600×*g*, 5 min). Lymphocytes were isolated from the pellet.

## Lung

Lung tissue was minced to 1 mm pieces using a scalpel and incubated in RPMI containing 1% BSA, 15 mM HEPES and 300 U/ml of Collagenase VIII (Sigma-Aldrich, St Louis, MO) in a 37 °C shaking incubator for ~30 min, pipetting to break up tissue halfway through. Lymphocytes were isolated from the single-cell suspension by gradient centrifugation with Ficoll-Hypaque (GE-Healthcare, 600 × *g*, 20 min).

## Skin

Ears were harvested and stored on ice in PBS/BSA. Ears were mechanically split, finely minced, and digested in RPMI with BSA, collagenase D (Roche), and Liberase TM (Roche) at 37 °C for 80 min. Leukocytes were separated by gradient centrifugation with Lymphoprep (StemCell Technologies, Inc.).

## Lymphoid tissues

Lymphoid tissues (spleen and lymph nodes) were isolated from surrounding tissues with tweezers and maintained on ice in PBS/BSA. Tissues were processed to a single-cell suspension by maceration through a 70 μm mesh. Spleen samples were then incubated with 1 ml ACK lysis buffer for 3 min to lyse red blood cells.

## RNA-sequencing

5 pools of 4 B6 mice were sorted for equal cell numbers (300) for CD127 + DN, MAIT, NKT, CD127 + CD4. 5 pools of 2 Cbir Rag−/− mice were sorted for Cbir DN population. Cells for RNA-sequencing were isolated from tissue as described above and sorted with a FACSAria III directly into 300 μl RLT (Qiagen). RNA was isolated using the RNAeasy micro kit (Qiagen). Quality control, library prep, and sequencing were performed at the Wellcome Trust Centre for Human Genetics, Oxford Genomics Centre using the SmartSeq2 protocol.

To obtain human data, CD4 T cells from three donors were sorted on a BD ARIA III sorter based on the expression of CD161 and CD56 as CD161[-], CD161[int], CD161[hi]CD56[-] or CD161[hi] CD56[+] cells. For each donor and cell population a total of 800 cell was sorted infour batches (200 cells each) directly in PCR tubes containing 4 μl SmartSeq2 lysis buffer. Reverse transcription was done directly from the sorted, lysed cells following the protocol published by Picelli et al.[78] in the MRC WIMM Sequencing Facility. cDNA libraries cDNA libraries were processed with a Nextera XT kit, using 8 bp barcodes, and were sequenced on a NextSeq500 sequencer.

**Tissue qPCR.** 3 mm pieces of colonic tissue were placed into RNA*later* (Qiagen) directly after sacrifice and stored at −20° Celsius until use. RNA was isolated using the RNAeasy mini kit (Qiagen) according to

manufacturer's instructions. cDNA was synthesized using the Super-script III reverse transcription kit (Life Technologies). Quantitative real-time PCR for the candidate genes was performed using the Taqman system in duplicate and presented relative to *Hprt*. All probes were obtained from Life Technologies: *Cd3e* (Mm01179194_m1), *Zbtb16* (Mm01176868_m1), *Zbtb7b* (Mm00784709_s1), *Il22* (Mm01226722_m1), *Il17a* (Mm00439618_m1), *Csf2* (Mm01290062_m1), *Ifng* (Mm01168134_m1), *Areg* (Mm00437583_m1), *Tnf* (Mm00443258_m1), *Il1b* (Mm01336189_m1), *IL12a* (Mm00434165_m1), *IL17f* (Mm00521423_m1), *Il10* (Mm00439614_m1).

## DSS colitis
Mice were pre-treated with a depleting dose of anti-CD4 antibody (GK1.5, BioXCell Cat # BE0003-1, 0.5–1 mg tested by batch) three days prior to DSS treatment and every 7 days for the duration of the experiment. 1.5% dextran sulfate sodium salt (36,000–50,000 M Wt, colitis grade, MP biomedical) was given in drinking water from day 0–5. Mice were weighed and monitored daily. Mice were culled and tissues harvested on day 19.

## Statistics and reproducibility
No statistical method was used to predetermine sample sizes. Data were collected and handled in either GraphPad 9 or in R (Version 4.0.3). Statistical analysis: repeated measures ANOVA with Dunnett's multiple comparisons test, Friedman tests with Dunn's multiple comparisons tests, Mixed-effects analysis with Tuckey's or Dunnett's multiple comparisons test, two-tailed Mann-Whitney test, 2-way ANOVA with Sidak's multiple comparisons test, Kruskal-Wallis test with Dunn's multiple comparisons test and regular one-way ANOVAs were computed in GraphPad 9. All tests in R were chosen after the normality of the datasets in this study was tested in GraphPad 9 using both, a Shapiro-Wilk and a Kolmogorov-Smirnov normality test. One-tailed Fisher's exact test scheme, GSEA p-values calulation using an adaptive multi-level split Monte Carlo scheme and Walds test to identify differentially expressed genes were computed in R. All R-based calculations were adjusted for multiple testing by the Benjamini-Hochberg method. All tests were two-tailed unless speci-fically stated otherwise. P-values smaller than 0.05 were considered significant. The *n* numbers for each experiment and the number of experiments involving mice are reported in each figure legend. Human samples were processed on an individual basis as they were coming in and hence, each datapoint represents a separate experi-ment. No data were excluded. Experiments were not randomized and investigators were not blinded to allocation during experiments and outcome assessment.

## Reporting summary
Further information on research design is available in the Nature Portfolio Reporting Summary linked to this article.

# Data availability
All RNA-sequencing datasets are available via permanent link as fol-lows: the human RNA-sequencing data generated in this study have been deposited in the EMBL-EBI ArrayExpress database under accession code E-MTAB-11440. The mouse RNA-sequencing data generated in this study have been deposited in the EMBL-EBI ArrayExpress database under accession code E-MTAB-11397. The Rhapsody single-cell data generated in this study have been depos-ited in the Gene Expression Omnibus database under accession code GSE207159. Source data are provided with this paper.

# Code availability
R scripts allowing the reproduction of the data can be obtained from: https://github.com/saxifragus-oxf/TMIC-project.

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

## Acknowledgements

C.-P.H. was supported by a Research Fellowship (403193363) from the Deutsche Forschungsgemeinschaft (DFG, 2018-2020), the Wellcome Trust (222426/Z/21/Z, awarded to P.K.) and a Medical Sciences Internal Fund (Pump-Priming: award 0009784). H.H.U. acknowledges research support from the Helmsley Charitable Trust and the NIHR Biomedical Research Centre Oxford. F.P. was supported by Wellcome Trust (095688/Z/11/Z and 212240/Z/18/Z). P.K. was supported by the Wellcome Trust (222426/Z/21/Z), the National Institute for Health Research (NIHR) Biomedical Research Centre (BRC), an NIHR Senior Fellowship, and the National Institutes of Health (NIH, U19 AI082630). E.T. was supported by Wellcome Trust (095688/Z/11/Z and 212240/Z/18/Z, awarded to F.P.), Nuffield Department of Medicine, MRC core grant reference MC_UU_00008, and the University of Oxford COVID Rebuilding Research Momentum Fund (CRRMF). We would like to thank Luke Barker for supporting animal husbandry and experimentation throughout the project. We thank Charles O. Elson III for the Cbir1 mouse strain, Dan Littman for the Hh7-2 mouse strain, and Dan Cua for the IL-23RGFP mouse strain. We want to thank Dr. Ida Parisi, Dr. Bryony Stott, and Miss Rhiannon Cook in the Kennedy Institute of Rheumatology Histology Service for tissue processing and staining. We thank the Oxford Genomics Centre at the Wellcome Centre for Human Genetics (funded by Wellcome Trust grant reference 203141/Z/16/Z) for the generation and initial processing of sequencing data. The following reagent(s) was/were obtained through the NIH Tetramer Core Facility: CD1d and MR1 tetramers. The MR1 tetramer technology was developed jointly by Dr. James McCluskey, Dr. Jamie Rossjohn, and Dr. David Fairlie, and the material was produced by the NIH Tetramer Core Facility as permitted to be distributed by the University of Melbourne. We want to thank Helen Ferry for flow cytometry panel design and cell sorting at the EMD/TGU flow cytometry facility and Neil Ashley for RNA-sequencing at the WIMM Single Cell Genomics Facility conducted as part of a Human Immune Discovery Initiative (HIDI) project awarded to C.-P.H. We acknowledge the contribution of the Oxford BRC Gastrointestinal Biobank supported by the NIHR BRC. The views expressed are those of the authors and not necessarily those of the NHS, the NIHR, or the Department of Health. We thank members of the Oxford TGU biobank, especially J. Chivenga, A. Isherwood, R. Williams, and M.

Cabrita for facilitating the collection of patient samples. We further acknowledge the contribution of the members of the Oxford IBD Investigators consortium: Dr. Carolina Arancibia, Dr. Adam Bailey, Professor Ellie Barnes, Dr. Noor Bekkali, Dr. Elizabeth Bird-Lieberman, Dr. Oliver Brain, Dr. Barbara Braden, Dr. Jane Collier, Professor James East, Dr. Lucy Howarth, Professor Paul Klenerman, Professor Simon Leedham, Dr. Rebecca Palmer, Dr. Fiona Powrie, Dr. Astor Rodrigues, Dr. Francesca Saffioti, Professor Alison Simmons, Dr. Peter Sullivan, Professor Holm Uhlig, Professor Jack Satsangi, Dr. Philip Allan, Dr. Timothy Ambrose, Dr. Jan Bornschein, Dr. Jeremy Cobbold, Dr. Emma Culver, Dr. Michael Pavlides, and Dr. Alissa Walsh. This work benefited from data collected by the ImmGen consortium[55].

## Author contributions

C.-P.H. contributed to conceptualization, data curation, formal analysis, investigation, methodology, project administration, validation, visualization, writing–original draft, review, and editing. D.C. contributed to investigation, validation, formal analysis, and writing-review, and editing. L.D. contributed to investigation, validation, and formal analysis. C.P. provided conceptualization, resources, investigation, project administration, and writing-review and editing. S.B. contributed to investigation, project administration, and writing—review and editing. N.I. contributed to data curation, formal analysis, and writing—review. H.A.D. contributed to investigation. Y.G. contributed to investigation and writing-review and editing. M.E.B.F. contributed to investigation and writing—review and editing. O.J.H. contributed to investigation, resources, and writing-review and editing. L.C.G. contributed to investigation, methodology, and writing—review and editing. E.H.M. contributed to the investigation and writing—review and editing. S.P. contributed to investigation. M.F. contributed to investigation and writing—review and editing. N.M.P. contributed to supervision and writing—review and editing. H.U. contributed to supervision—review, and editing. E.M. contributed to methodology, validation, and visualization. F.P. contributed to conceptualization, funding acquisition, supervision, validation, and writing-review and editing. P.K. contributed to conceptualization, funding acquisition, supervision, validation, and writing-review and editing. E.T. contributed to conceptualization, data curation, formal analysis, investigation, methodology, project administration, supervision, validation, visualization, writing-original draft, review, and editing.

## Competing interests

F.P. received consultancy or research support from GSK, Novartis, Janssen, Genentech, and Roche. H.H.U. has received research support or consultancy fees from Janssen, UCB Pharma, Eli Lilly, AbbVie, Celgene, OMass, and MiroBio. P.K. has done consultancy for UCB and Medimab. The remaining authors declare no competing interests.
