## [Peer Review File · Nature Communications]

A conserved population of MHC II-restricted, innate-like, commensal-reactive T cells in the gut of humans and miceREVIEWER COMMENTS

Reviewer #1 (Remarks to the Author):

In this work, Thornton et al explore a population of T cells, which they call “T-MHC-II restricted, innate-like, commensal-reactive” (TMIC), found in the gut of mice and humans, that react quickly to microbial stimulation by producing inflammatory cytokines, seem to be MHCII restricted, and have other features of innate like lymphocytes. These T cells express high levels of CD161, similar to most human colonic CD4 T cells, and most express CD56 (although in mice they are CD4/CD8 double negative). They have many of the characteristics of innate like T cells in being able to rapidly respond by producing cytokines in response to inflammatory cytokines IL12/IL18 and IL23, and transcriptomically resemble such cells. The authors extend their analysis to human colitis, and murine models of colitis, as well as T cell receptor transgenic mouse models. The work is quite comprehensive in its characterization of these cells, although there are some concerns about naming these a unique population of cells given the diverse T cell population that reside in the gut that are reactive to microbial productions. Some concerns are detailed below:

1. The authors attempt to exclude innate like cells such as MAIT, and invariant NKT cells, as well as gamma delta T cells, however, there are populations of non-conventional NKT cells that may be included in their analysis and it's not clear whether these can be excluded.
2. It seems that the cells used for the analysis of cytokine production represents a mixed population that includes cells that are presenting antigen via MHCII, as well as the T cells. Do the colonic T cells respond to supernatants from T cell depleted non-colonic T cells previously exposed to these microbial products? In other words, do the heat killed microbes trigger non-T cell production of soluble factors that work to enhance or co-activate the T cells in question to produce cytokine?
3. In referring to the data in supplemental figure 1E, the authors state that IL23 is able to induce IFN γ and granzyme, however looking at the data, this does not seem to be statistically significant.
4. The authors state that “that the acquisition of the effector profile and innate-like features in CD161hi CD4 T cells represents a unique feature of microbe-reactive CD4 T cells occurring independently of the formation of TRM cells.” However can the authors rule out that these cells being studied by the authors represent a type of TRM cells.
5. The authors indicate that these cells are found at mucosal surfaces. Are they found in significant numbers in the lung, particularly in the mouse models (aside from the analysis shown in figure 3)?
6. In the data shown in figure 3 characterizing the Cbir1 T cell receptor transgenic mice, it would be useful to compare the colonic T cells, with T cells in the periphery. Similarly, while the authors make the argument that the phenotype of the cells develop in situ, it would be useful to compare their phenotypes in the thymus to underline this evidence.
7. While this may not be possible, it would be of interest to determine if the phenotype of the T cells in the Hh-TCR transgenic mice would develop this phenotype if colonized with *Helicobacter hepaticus*.
8. The authors write that “..that cells with a CD161hi phenotype are present in equal or even slightly elevated numbers per gram of tissue compared to samples obtained from non-inflamed or normal tissue..”, and “..Similar trends were also found in two mouse models of colitis..” in reference to supplemental figure 6. However, this isn't clear as in supplemental figure 6a, there is no apparent difference in the populations depicted. Also, are the numbers of these cells in the mouse model different?
9. The data analyzing the DSS-induced colitis mouse model shown in figure 5 (c-g) are bit

confusing. The data in figure 5c does not look like there is a significant difference in the weight loss between CD4 depleted Rag deficient mice and CD4-depleted Cbir mice. Is there a statistically significant difference in the weight loss curves? What about in Cbir mice where CD4 T cells have not been depleted?

10. Also in reference to that same set of experiments, figure 5g is referred to as spleen, but these sections look like sections of intestine. Is that correct? Was there an increase in IFN γ and IL17A colonic tissue as well?

11. In supplemental figure 7d, what is the gating progression that is being shown (i.e. previous gates that lead to these graphs)?

12. The CFSE dilution experiment shown in figure 3F is a bit difficult to discern between the different conditions. Perhaps these histograms can be offset?

13. Similarly, the color scheme used in the depiction of the RNA-sequencing analysis by PCA makes it a bit difficult to discern the different cells (particularly figure 4 and supplemental figure 4D). It seems that there are some cells missing, or they may be difficult to see given the similarity of colors used.

14. There is a mismatched reference on line 313 that should be corrected.

Reviewer #2 (Remarks to the Author):

In this study Thornton et al. analyzed in humans and mice gut ab T cells recognizing microbial antigens. In humans, low frequency (0.2%) of lamina propria CD4 cells can be activated by E coli, S aureus or C albicans and produced TNF α . They are CD161^{high} and CD56^{+/-} cells. CD161^{high} CD4 T cells expressed IL-18R, PLZF. In mice, they described IL-23R-GFP⁺ abT cells in colon lamina propria, that are IL-7R⁺, IL-18R⁺, and are more abundant in colon, skin and kidney than in lymphoid tissues. Higher frequency of these cells expressed IL-7R in older mice and a lower frequency in germ free mice suggesting that microbiota could impact their expansion/maturation. To further study these cells in mouse model, they used Cbir1 TCR Tg RAGKO mice, this TCR recognizing a microbial Ag. Transcriptomic analysis shows shared transcription profiles between human and mouse lamina propria abT cells (CD161^{high} in humans and Cbir1 T cells in mice). Finally, the authors show increased frequency of CD161^{high} abT cells in biopsies from inflamed UC compared to non-inflamed and performed mouse experiment to address their function in DSS-induced colitis.

While some of the results are interesting, there are several concerns about this study. The link between microbial Ag specificity and the phenotype/function of these abT cells is not fully established and further experiments are required.

Major concern:

The in vivo data in Figure 6 shown to demonstrate the potential role of gut DN abT cells (called TMIC) are based on Cbir1 TCR Tg mice on RAGKO compared to RAGKO mice, that do not contain any T cells. This system does not seem appropriate to determine whether microbe reactive abT cells are playing a specific role in DSS induced colitis. Instead, the authors could purify IL-23R-GFP⁺ polyclonal abT cells (from B2M KO mice), which they characterized as mainly anti-microbial reactive T cells and transfer them into recipient mice, as compared to recipient mice receiving IL-23R-GFP negative abT cells.

It would be interesting/important to analyze the microbial reactivity of CD161^{high} abT cells in biopsies from inflamed UC, as they do in Figure 1. Indeed, there is no evidence that all CD161^{high} abT cells are reactive toward bacterial Ag. It is possible that conventional abT

cells upregulated CD161 in an inflammatory context. In vitro response to fecal lysate in presence or not of blocking class II would add a lot to this study.

Other comments:

Figure 1: Would be important to see the staining on bacteria reactive abT cells for PLZF, IL-18R, IFNg....

Figure 1m: there is no anti-MHC class II blocking experiment with *C albicans* and the data with *S aureus* should be strengthened (may be one outlier sample is making the difference).

As compared to previous studies on mucosal T cells, it is unexpected to don't see expression of tissue repair genes in these TMIC cells in homeostasis situation in human and mice. What about TGF-B expression, amphiregulin.... Such proteins and gene expression could be analyzed in both human and mouse TMIC, as they defined them.

Reviewer #3 (Remarks to the Author):

In their paper, Thornton et al, study T cells in human and mouse gut and find a population of human CD4 T cells with high expression of CD161, and in mouse a CD4/CD8 DN T cell subset, sharing similar innate-like transcriptional profiles. The findings are interesting but fail to fully convince regarding key aspects regarding the claim that this is a new cell subset with distinct characteristics. This reviewer has the following concerns and comments:

In figure 1A, low frequencies of antigen-stimulated TNF are shown in colonic CD4 T cells. It will be important to show the gating strategy. Were MAIT cells gated out from this dataset? If not, the most straight forward interpretation would be that these low frequency responses are CD4+ MAIT cells (all three tested microbes give rise to MAIT antigens). This gating out of MAIT cells seems to be done later in the figure 1 panels F and G on cytokine stim, so this should be done for the antigen stim as well. In fact, for the cytokine stim it is not as necessary.

Also, in figure 1A the incubation time according to description in mat and met is 2+6 hrs. This is relatively short if you feed whole bugs to a mix of mononuclear cells, just 2 hrs for uptake and processing. Would responses be stronger with say 4+6 hrs?

The authors want to call the CD161hi CD4 T cells "Tmic", where MIC stands for MHC II-restricted, innate-like, commensal-reactive. Innate-like is relatively well supported. There is not much data to support the claim that this population is MHC II-restricted and commensal-reactive.

The authors show partial blocking of TNF production in response to microbes by anti-MHC class II mAb. It would be important to test MR1 and CD1 blocking as well to be able to more firmly say that these cells are not MAIT or NKT cells.

The ability of CD4+ CD161hi cells to produce IL-17 and IL-22 is very limited and oversold in the manuscript (supplementary figure 1E).

Is PLZF expression in the CD4mic T cell comparable to MAIT cells?

The authors aim to suggest that there is a mouse DN T cell population with similar characteristics. However, they do not really address if there is a similar DN T cell population in humans?

The mouse DN T cells and human CD161^{hi} CD4 T cells seem to have similar transcriptional profiles. But is unclear if the mouse DN T cells also contain the same low frequencies of cells reactive to *E. coli*, *S. aureus*, and *C. albicans*?

Do the mouse DN T cells express innate markers such as NK1.1 or similar analogous to the human CD161 and CD56?

For figure 5A the same concern exists as for figure 1A, are the CD161⁺ CD4 T cell CD4⁺ MAIT cells? Also, the increase in absolute counts of CD161^{hi} cells seem less than for the overall CD4s. Does this mean the CD161^{hi} population decreases as a percentage?

In several figures there are examples of "representative" FACs plots and histograms and there is no mentioning of out of how many independent experiments.

We thank the reviewers for their thoughtful contributions. We have added data to support our claim that the cells described in this study are MHCII-, innate-like, and commensal-reactive and hope that we have addressed the reviewers' concerns in the point-by-point response below and changes to the manuscript.

Reviewer #1 (Remarks to the Author):

In this work, Thornton et al explore a population of T cells, which they call "T-MHC-II restricted, innate-like, commensal-reactive" (TMIC), found in the gut of mice and humans, that react quickly to microbial stimulation by producing inflammatory cytokines, seem to be MHCII restricted, and have other features of innate like lymphocytes. These T cells express high levels of CD161, similar to most human colonic CD4 T cells, and most express CD56 (although in mice they are CD4/CD8 double negative). They have many of the characteristics of innate like T cells in being able to rapidly respond by producing cytokines in response to inflammatory cytokines IL12/IL18 and IL23, and transcriptomically resemble such cells. The authors extend their analysis to human colitis, and murine models of colitis, as well as T cell receptor transgenic mouse models. The work is quite comprehensive in its characterization of these cells, although there are some concerns about naming these a unique population of cells given the diverse T cell population that reside in the gut that are reactive to microbial productions. Some concerns are detailed below:

1. The authors attempt to exclude innate like cells such as MAIT, and invariant NKT cells, as well as gamma delta T cells, however, there are populations of non-conventional NKT cells that may be included in their analysis and it's not clear whether these can be excluded.

The reviewer is correct, non-conventional (Type II) NKT (and possible also MAIT) cells do exist and wouldn't be excluded by the commonly used MRI- and CD1d-tetramers or the TCR-antibodies used in the human experiments.

To address this issue, we repeated our microbial-response assay and added blocking antibodies targeting MRI or CD1d to it. As shown in our revised Figure 1, I, blocking of either these molecule in contrast to aMHCII-treatment did not have a significant impact on the antimicrobial response suggesting that neither MRI nor CD1d are restricting elements for TMICs. While type II NKTs used a diverse range of TCRs and are hard to study or exclude that way, CD1d-restriction represents a defining feature of them. Therefore, our results indicate that type II NKTs do not contribute to the antimicrobial responses we are describing.

2. It seems that the cells used for the analysis of cytokine production represents a mixed population that includes cells that are presenting antigen via MHCII, as well as the T cells. Do the colonic T cells respond to supernatants from T cell depleted non-colonic T cells previously exposed to these microbial products? In other words, do the heat killed microbes trigger non-T cell production of soluble factors that work to enhance or co-activate the T cells in question to produce cytokine?

The reviewer makes an interesting point. Given that TMICs express several cytokine-receptors and comparatively high levels, cytokine-triggered responses are definitely possible.

In fact, this was one of the reasons why an 8 hour incubation was chosen for the microbe-response assay as this precedes the timing for purely cytokine-driven responses in MAIT cells.

There are some obvious assays that could be done to give a definitive answer the reviewers questions in order to rule any non-contact dependent responses out. E.g. as the reviewer suggested one could separate the T cells and non- T cells via sorting or bead-based separation and then do supernatant-transfers or transwell assays in order to determine if T cell activation still takes place.

There were however a couple of technical problems preventing us from addressing the reviewers concern that way:

- To address a question brought forward by reviewer 2, we performed a single cell sequencing experiment that involved cell sorting of gut-derived cells. In this experiment we saw that the viability of the cells from our samples deteriorated quickly upon sorting rendering that method difficult to employ for any assay that would require subsequent culturing.
- Particularly the myeloid cell compartment, which is presumably quite important for antigen presentation, seemed to be affected quite badly.
- Sufficient cell numbers are an issue. All three reviewers requested additional experiments involving human colonic cells. These included repeats of the microbe-response assay previously shown in the manuscript including additional blocking conditions (aCD1d, aMR1). We agree with the reviewers that these experiments are very important and prioritized them. Unfortunately, after setting those up there is not enough material left to isolate sufficient cell numbers for supernatant-transfers or transwell assays.

To still address this reviewers concerns though, we would like to point the following things out:

- TNF expression by CD161hi CD4 T cells was only observed upon TCR stimulation in all our experiments involving comparisons between cytokine-stimulation (mainly IL12/18) and TCR-stimulation (via plate-bound aCD3 + soluble aCD28). This was true for either classic FACS-experiments (here gated on iTCR-CD161hiCD4+):

And also seen on the transcript level in our Rhapsody single-cell experiment:

(cytokine = IL12/18, cells had been stimulated overnight before sort, single cell capture, library prep and sequencing).

Based on these observations and similar to what has been shown for MAIT cells stimulated with E.coli in vitro, we consider it very likely that TNF in TMICs is also a TCR-driven effector molecule.

- *In line with that idea, TNF was used by others to specifically detect microbe-responsive cells in the gut before: Hegazy et al.: Circulating and Tissue-Resident CD4⁺ T Cells With Reactivity to Intestinal Microbiota Are Abundant in Healthy Individuals and Function Is Altered During Inflammation, Gastroenterology, 2017.*
- *We show that microbe-induced TNF-production by colonic CD4s can be blocked by aMHCII antibodies which strongly suggests a contact-dependent mechanism. In our revised manuscript we have confirmed and expanded these data (revised Figure 1, I).*
- *We did previously show that IL-12, IL-18 and to some extent IL-23 can induced TCR-independent responses in TMICs. To test whether these contribute to the TNF-production shown, we added blocking neutralizing antibodies against IL18 and IL12p23 to our assay. As shown in our revised Figure 1, I, blocking of these cytokines had no significant impact on TNF production, suggesting that these are not required*

Based on previous results from experimental models using MAIT cells, from the new data we obtained from two different systems we used to assess TNF-production in human colonic CD4s as well as neutralization experiments, we conclude that there is very strong evidence for a contact-based activation.

3. In referring to the data in supplemental figure 1E, the authors state that IL23 is able to induce IFN γ and granzyme, however looking at the data, this does not seem to be statistically significant.

There clearly is a visible upregulation of both IFN γ and GzmB as shown in the FACS plots, but we agree that the impact of IL-23 stimulation is relatively small compared to the one seen with IL-12/18 or aCD3 and does not reach statistical significance after correction for multiple testing. As these results do not change the message of our manuscript, we re-worded the corresponding text sections to avoid overstating the importance of these findings.

4. The authors state that “that the acquisition of the effector profile and innate-like features in CD161hi CD4 T cells represents a unique feature of microbe-reactive CD4 T cells occurring independently of the formation of TRM cells.” However can the authors rule out that these cells being studied by the authors represent a type of TRM cells.

We agree with the reviewer. Our data and analysis show that the specific TRM-associated gene signatures don't correlate very well with the TMIC gene signature, however the TRM signatures published by Beura et al. and Milner et al. were obtained from cells from specific models and tissues and hence might not perfectly represent all TRMs in all settings. We re-phrased the section in the text to take this into account.

5. The authors indicate that these cells are found at mucosal surfaces. Are they found in significant numbers in the lung, particularly in the mouse models (aside from the analysis shown in figure 3)?

We thank the reviewer for pointing out this interesting aspect of Tmic biology. Indeed, the cells are found at similarly high frequency in the lung as in the colon. This data is now included in Fig 2 C and D for B6 and expanded data for Cbir in Fig 3C and D.

6. In the data shown in figure 3 characterizing the Cbir1 T cell receptor transgenic mice, it would be useful to compare the colonic T cells, with T cells in the periphery. Similarly, while the authors make the argument that the phenotype of the cells develop in situ, it would be useful to compare their phenotypes in the thymus to underline this evidence.

This is a very good point. We have now included a more comprehensive description of the antigen specific cells in the periphery of Cbir mice. FACS of the thymus shows a small but appreciable PLZF+ IL7R+ population in the Tmic gate in B6 and Cbir mice (Supp fig 4E). This suggests that these cells may develop in the thymus and then engraft in mucosal sites similar to other innate T cells.

7. While this may not be possible, it would be of interest to determine if the phenotype of the T cells in the Hh-TCR transgenic mice would develop this phenotype if colonized with *Helicobacter hepaticus*.

*This is a very interesting suggestion. We infected some Hh-TCR transgenics as the reviewer suggested. Despite overt colitis, the TCR transgenic cells did not develop the Tmic phenotype. There are several reasons this may be the case. It is possible that the time of colonization is important, and the niche that would be occupied by Tmics during development is filled by another cell type. The affinity of the TCR in the thymus may be important for differentiation down this path (new data showing cells in the thymus may support this conclusion). Additionally, *Helicobacter hepaticus* is a pathobiont instead of a bona fide commensal so may favor a more classical CD4 T cell response. While we think these are interesting prospects, we believe that addressing these hypotheses is outside the scope of this study.*

8. The authors write that “..that cells with a CD161hi phenotype are present in equal or even slightly elevated numbers per gram of tissue compared to samples obtained from non-inflamed or normal tissue..”, and “..Similar trends were also found in two mouse models of colitis..” in reference to supplemental figure 6. However, this isn’t clear as in supplemental figure 6a, there is no apparent difference in the populations depicted. Also, are the numbers of these cells in the mouse model different?

The reviewer is correct to point out this discrepancy. We wanted to make the point that the Tmics persist in the context of inflammation and have now clarified this in the text.

9. The data analyzing the DSS-induced colitis mouse model shown in figure 5 (c-g) are bit confusing. The data in figure 5c does not look like there is a significant difference in the weight loss between CD4 depleted Rag deficient mice and CD4-depleted Crib mice. Is there a statistically significant difference in the weight loss curves? What about in Cbir mice where CD4 T cells have not been depleted?

This is an important observation. Because we studied the mice through peak weight loss and through weight recovery, we would not expect there to be a statistical difference between the curves. They overlap before and after peak weight loss. This is why we quantified peak weight loss. Due to animal regulations in the UK, we are not permitted to induce more extreme weight loss, which might show more statistical differences.

10. Also in reference to that same set of experiments, figure 5g is referred to as spleen, but these sections look like sections of intestine. Is that correct? Was there an increase in IFN γ and IL17A colonic tissue as well?

We thank the reviewer for finding this error. Fig 5E refers to the spleen weight, and the rest of the figure is focused on the colon tissue. We have included IFN γ at the reviewer’s request, but IL17a was not reproducibly amplified in our assay so we are not able to make conclusions about its expression.

11. In supplemental figure 7d, what is the gating progression that is being shown (i.e previous gates that lead to these graphs)?

The gating strategy has now been shown in full (Supplemental Figure 8, D in the revised manuscript).

12. The CFSE dilution experiment shown in figure 3F is a bit difficult to discern between the different conditions. Perhaps these histograms can be offset?

Very good point. The experiment has now been replaced by a cleaner experiment that includes vancomycin treated faeces as an additional control with offset plots.

13. Similarly, the color scheme used depiction of the RNA-sequencing analysis by PCA makes it a bit difficult to discern the different cells (particularly figure 4 and supplemental figure 4D). It seems that there are some cells missing, or they may be difficult to see given the similarity of colors used.

We double-checked and all cell types are there.

Given the number of different populations, esp. in the main figure, we agree that it is difficult to differentiate all of them. We changed the coloring of all populations on the main and supplemental figure (Supp Figure 5 in the revised manuscript) to address this and hope the new color-scheme makes it easier to spot the relevant cell populations.

14. There is a mismatched reference on line 313 that should be corrected.

We thank the reviewer for pointing this mistake out. The proper reference was added.

Reviewer #2 (Remarks to the Author):

In this study Thornton et al. analyzed in humans and mice gut ab T cells recognizing microbial antigens. In humans, low frequency (0.2%) of lamina propria CD4 cells can be activated by E coli, S aureus or C albicans and produced TNFa. They are CD161^{high} and CD56^{+/-} cells. CD161^{high} CD4 T cells expressed IL-18R, PLZF. In mice, they described IL-23R-GFP⁺ abT cells in colon lamina propria, that are IL-7R⁺, IL-18R⁺, and are more abundant in colon, skin and kidney than in lymphoid tissues. Higher frequency of these cells expressed IL-7R in older mice and a lower frequency in germ free mice suggesting that microbiota could impact their expansion/maturation. To further study these cells in mouse model, they used Cbir1 TCR Tg RAGKO mice, this TCR recognizing a microbial Ag. Transcriptomic analysis shows shared transcription profiles between human and mouse lamina propria abT cells (CD161^{high} in humans and Cbir1 T cells in mice). Finally, the authors show increased frequency of CD161^{high} abT cells in biopsies from inflamed UC compared to non-inflamed and performed mouse experiment to address their function in DSS-induced colitis.

While some of the results are interesting, there are several concerns about this study. The link between microbial Ag specificity and the phenotype/function of these abT cells is not fully established and further experiments are required.

Major concern:

1. The in vivo data in Figure 6 shown to demonstrate the potential role of gut DN abT cells (called TMIC) are based on Cbir1 TCR Tg mice on RAGKO compared to RAGKO mice, that do not contain any T cells. This system does not seem appropriate to determine whether microbe reactive abT cells are playing a specific role in DSS induced colitis. Instead, the authors could purify IL-23R-GFP⁺ polyclonal abT cells (from B2M KO mice), which they characterized as mainly anti-microbial reactive T cells and transfer them into recipient mice, as compared to recipient mice receiving IL-23R-GFP negative abT cells.

We thank the reviewer for these comments. It would be ideal to have an optimized cell transfer system, but as summarized in Supp Fig 8A-C, we were not able to reconstitute the colon with transferred cells. This is likely due to the strong tissue resident signature of these

cells. We believe the Cbir1 system comparing the presence and absence of cells with this phenotype is sufficient to show what these cells are capable of in vivo. Increasing evidence supports overlapping functions of innate-like T cells, which is likely to be the case with Tmic cells so focusing on a reductionist system was the cleanest way to look for a function in vivo.

It would be interesting/important to analyze the microbial reactivity of CD161^{high} abT cells in biopsies from inflamed UC, as they do in Figure 1. Indeed, there is no evidence that all CD161^{high} abT cells are reactive toward bacterial Ag. It is possible that conventional abT cells upregulated CD161 in an inflammatory context. In vitro response to fecal lysate in presence or not of blocking class II would add a lot to this study.

The reviewer raised a couple of interesting questions: In agreement with our findings, Hegazy et al. previously showed that the total frequencies of microbe-reactive CD4 T cells stay constant or tend to be slightly higher in IBD patients (CD and UC, Hegazy, West et al, Gastroenterology 2017). Importantly though, apart from cytokine-expression, the microbe-reactive cells were not phenotyped (e.g. for CD161) in more detail in that study. We agree with the reviewer that additional studies in the inflamed tissue would be interesting to further analyze the regulation of CD161-expression and microbe-responsiveness. Unfortunately, the cell numbers we can isolate from the biopsies currently available to us are not sufficient to perform these experiments in the foreseeable future: we regularly obtain 0.5-3 million LPMCs from a set of biopsies and the microbial response assay, due to the low % of responding cell against any given microbe, requires a million cells/condition. Future studies using larger samples, e.g. from resection could clarify this.

Given the strong correlation between microbe-responsiveness and CD161 expression we observed in the normal tissue, the data from the murine colitis models showing persistence of TMICs in the inflamed colon and MHCII-dependent response to fecal lysates (revised Figure 3, I) and the fact that UC has been linked, at least in part, to overzealous anti-microbial adaptive immune responses by various groups in the past, we think there is good evidence that TMICs would also persist and would be relevant in humans.

Other comments:

2. Figure 1: Would be important to see the staining on bacteria reactive abT cells for PLZF, IL-18R, IFN γ ...

We thank the reviewer for this suggestion. We repeated our microbe-response assay to characterize the TNF⁺ cell population in more detail and included the results in our revised figure 1. IL18Ra was expressed by almost all responding cells and in a significantly higher frequency than found on colonic CD4 T cells as a whole. IFN γ -expression however differed between the different microbes: about 40% of the E.coil-responsive cells co-expressed IFN γ , while fewer S.aureus- or C.albicans-specific cells did so.

PLZF staining was higher in microbe-responsive cells compared to non-responding CD4s and importantly was also higher compared to cells producing TNF in response to SEB-exposure, which was used as a positive control in the assay. The latter suggests that the level of PLZF observed here is not just a result of the activation of cells and reflects a more intrinsic property of microbe-reactive cells in the colon.

3. Figure 1m: there is no anti-MHC class II blocking experiment with *C. albicans* and the data with *S. aureus* should be strengthened (may be one outlier sample is making the difference).

Again, we thank the reviewer for this comment. We repeated the assay several times and included C.albicans (as well as blocking against other antigen-presenting molecules). Results are displayed in our revised Figure 1 I.

Since the magnitude of the antimicrobial responses differed considerably between donors and microbes, we decided to change the way how these data are displayed: instead of showing total cell numbers, we are now displaying the response measured upon MHCII-blockade as % of the response seen in the corresponding control (isotype-treated) sample. Our data clearly show the strong impact of MHCII-blocking on the responses against both S.aureus and C .albicans and we hope the reviewer agrees that the data are more accessible now.

4. As compared to previous studies on mucosal T cells, it is unexpected to don't see expression of tissue repair genes in these TMIC cells in homeostasis situation in human and mice. What about TGF-B expression, amphiregulin.... Such proteins and gene expression could be analyzed in both human and mouse TMIC, as they defined them.

Expression of tissue-repair-associated genes is indeed a feature all described innate-like T cells populations share.

Following the reviewer's suggestion, we included new experimental data to address this for TMICs in new Supplementary Figure 2, Fig 2e, and Fig 5i.

For the human cells we stimulated colonic CD4s with IL12/18, plate-bound aCD3 + soluble aCD28 or both overnight and then performed a single cell experiment using our Rhapsody Express platform. We did a targeted mRNA expression analysis using BDs Immune response panel as a base and added a custom panel including the tissue-repair associated genes published by the Belkaid group. To identify CD161hi CD4s, we did include Abseq-antibodies against CD161 and CD56, allowing us to identify the same cells as in our FACS analysis. The tissue repair list was previously used by the Klenerman and McCluskey groups to analyze tissue repair in murine and human MAIT cells and similar to the findings there, our analysis shows an upregulation of tissue-repair associated genes in human CD161hi CD4 T cells upon TCR, TCR+ cytokine but not cytokine-stimulation alone. Leading edge genes included well-known repair factors like AREG and VEGF. The mouse data is included in Fig 2, suggesting an increase with TCR engagement similar to the data from human Tmics. In contrast, in the context of colitis, we do not see an increase in Areg, suggesting that this pathways the dominant activation pathway of these cells in colitis is through cytokine signaling. We have now included discussion of this point in the main text.

Reviewer #3 (Remarks to the Author):

In their paper, Thornton et al, study T cells in human and mouse gut and find a population of human CD4 T cells with high expression of CD161, and in mouse a CD4/CD8 DN T cell subset, sharing similar innate-like transcriptional profiles. The findings are interesting but fail to fully convince regarding key aspects regarding the claim that this is a new cell subset with distinct characteristics. This reviewer has the following concerns and comments:

1. In figure 1A, low frequencies of antigen-stimulated TNF are shown in colonic CD4 T cells. It will be important to show the gating strategy. Were MAIT cells gated out from this dataset? If not, the most straight forward interpretation would be that these low frequency responses are CD4+ MAIT cells (all three tested microbes give rise to MAIT antigens). This gating out of MAIT cells seems to be done later in the figure 1 panels F and G on cytokine stim, so this should be done for the antigen stim as well. In fact, for the cytokine stim it is not as necessary.

We included a full gating in our revised Supplementary Figure 1, A.

The CD4 T cells we are displaying were gated as live, CD3+ and TCRgd+, Va7.2+ (including all MAIT and GEM T cells) and Va24-Ja18+ (iNKTs) as well as CD8+ cells were excluded and CD4+ cells were analysed.

We apologize that we didn't clearly communicate this in the previous version of the manuscript and amended the main text, figures and figure legends accordingly.

Regarding the possible contribution of MAIT cells to the response, we would also like to add that we repeated our microbe-response assay (to answer several other questions from the three reviewers) and included an aMRI blocking antibody. As shown in our revised Figure 1 I, blocking aMRI had no significant effect on the anti-microbial response and hence, we conclude that MAIT cells do not contribute to the responses we describe.

2. Also, in figure 1A the incubation time according to description in mat and met is 2+6 hrs. This is relatively short if you feed whole bugs to a mix of mononuclear cells, just 2 hrs for uptake and processing. Would responses be stronger with say 4+6 hrs?

The 8-hour protocol we used is based on the publication from Hegazy, West et al. from 2017.

When we set up the assay in our lab for this project we also tested longer incubation times, e.g. 16h. Below, we include a summary of these early experiments:

We did not see an increase of the % of TNF+ cells with longer incubation times. However, we noted that the background seemed to increase and hence decided to perform our experiments using the 8h protocol.

3. The authors want to call the CD161hi CD4 T cells “Tmic”, where MIC stands for MHC II-restricted, innate-like, commensal-reactive. Innate-like is relatively well supported. There is not much data to support the claim that this population is MHC II-restricted and commensal-reactive.

We appreciate the reviewer’s skepticism and appreciate the need to bring key points out early in the manuscript. Figure 1 now includes more data to support MHCII restriction and commensal-reactivity. We have also signposted the points that support the Tmic name as they come up in the text.

4. The authors show partial blocking of TNF production in response to microbes by anti-MHC class II mAb. It would be important to test MR1 and CD1 blocking as well to be able to more firmly say that these cells are not MAIT or NKT cells.

To address this issue we repeated our anti-microbe assay with several new samples and included blocking antibodies against MR1 and CD1d as the reviewer suggested. The new data are included in our revised Figure 1. In contrast to MHCII-blocking we could not detect a significant impact of either MR1- or CD1d-blockade on the TNF-production demonstrating that NKT and MAIT cells indeed do not contribute to the responses.

5. The ability of CD4+ CD161hi cells to produce IL-17 and IL-22 is very limited and oversold in the manuscript (supplementary figure 1E).

We agree that levels of these cytokines are much lower than e.g. the expression of IFN γ or GzmB observed in the same assay or in our IL12/18-experiments.

Taking the reviewers comment into account, we amended the main text to point out that these cytokines are all expressed by a comparatively low % of cells.

These low frequencies indeed were somewhat of a surprise given that previous publications (e.g. Kleinschek et al and Hegazy et al) suggested that human colonic CD4s are rich in cells with Th17-functionality. However, we would like to refer to our discussion and point out again that a lot of the existing data on human colonic CD4s were derived from IBD patients. As Th17 cells are important contributor to the disease, we hypothesize that the phenotype we found represents a more “normal” effector program.

6. Is PLZF expression in the CD4mic T cell comparable to MAIT cells?

We re-analyzed our data and compared the expression in TMICs to CD161hi Va7.2+ cells (note: these were almost exclusively CD8+ or DN) in the respective donors.

PLZF-expression in MAIT cells was notably higher than in TMICs, further supporting the notion that TMICs are not MAIT cells.

A summary of the analysis of MAIT-PLZF expression is included in the response to point 7. below (subfigure E).

7. The authors aim to suggest that there is a mouse DN T cell population with similar characteristics. However, they do not really address if there is a similar DN T cell population in humans?

We thank the reviewer for this comment. We did indeed considered this initially when we started the project. However, analysis of the small DN population that can be found in the human colon revealed:

- *gdTCR-Va7.2-Va24-Ja18- DN T cells in humans did not respond to any of the microbes we tested in any of the donors included in our study (subfigure A and B)*
- *only a subset of these cells did express IL18Ra and overall human DN cells were more comparable to CD161- CD4 T cells in that regard (subfigure C and D)*
- *PLZF-expression in human DN cells was on the same level as most CD4s and significantly lower than in CD161hi CD4 T cells (subfigure E)*

Based on these characteristics we concluded that human DN cells do not show any innate-like characteristics and do not participate in the response against the microbes we tested.

8. The mouse DN T cells and human CD161^{hi} CD4 T cells seem to have similar transcriptional profiles. But is unclear if the mouse DN T cells also contain the same low frequencies of cells reactive to *E. coli*, *S. aureus*, and *C. albicans*?

The reviewer makes a very good point. We would expect the mice to also have E. coli reactivity (potentially less S. aureus and C. albicans) due to the microbiome in our animal facility. We did attempt to study E. coli reactive cells from mouse colons; however, this proved very challenging due to low cell numbers in steady-state colons. This points to the

importance of having tools such as the Cbir TCR transgenic to study these relatively rare cell populations.

9. Do the mouse DN T cells express and innate markers such as NK1.1 or similar analogous to the human CD161 and CD56?

While mice do not have a CD161 analogue, we did stain for NK1.1 when studying Tmics in B6 and Cbir. In both cases, Tmics did not express NK1.1. In the DN population as a whole, CD127 and NK1.1 are nearly mutually exclusive. The B6 data has now been added to Fig 2B.

10. For figure 5A the same concern exists as for figure 1A, are the CD161+ CD4 T cell CD4+ MAIT cells? Also, the increase in absolute counts of CD161hi cells seem less than for the overall CD4s. Does this mean the CD161hi population decreases as a percentage?

As for the cells in Figure 1, cells expressing TCRg/d, Va7.2 and Va24-Ja18 were gated out and hence, no CD4+ MAIT cells were included in these analyses presented.

Again, we have to apologize for not labelling the figures clearly enough and have amended the main Figure, text and figure legend.

The reviewer is correct in the assumption about the % of CD161hi decreasing in IBD.

Earlier versions of the Figure included the % information:

Clearly showing the decreasing % of CD161hi and increasing % if CD161- cells particularly in inflamed UC tissue.

Looking at the total numbers, it is quite clear though that there is a massive increase of CD161- cells in IBD tissue. UC is characterized by T cell infiltration (in line with that we routinely were able to isolate considerable higher cell numbers from UC samples) and since the majority of peripheral CD4 T cells is CD161-, we believe that CD161hi cells in IBD get "diluted" by the infiltrating cells.

We presented and discussed these data in several internal seminars in Oxford previously and the feedback we got overall was that the % data were not considered to be helpful and - given

the known infiltration of cells in IBD – are somewhat misleading, as the intuitive interpretation would be that TMICs are decreased in IBD. Since total cell counts show that this is not the case, we would prefer to not include the %.

11. In several figures there are examples of “representative” FACs plots and histograms and there is no mentioning of out of how many independent experiments.

This was the case the experiments requiring fresh human resections tissue (e.g. all the microbe-responsiveness data). This assay has to be performed using freshly isolated cells as the myeloid cells required for antigen-presentation don not seem to survive freeze-thaw cycles very well. As human samples came in one-by-one each dot in these figures technically represents a separate experiment. We included this information in the Figure legends were appropriate.

REVIEWERS' COMMENTS

Reviewer #1 (Remarks to the Author):

The authors have addressed my major concerns, given the experimental restraints.

Reviewer #2 (Remarks to the Author):

This study describes of subset of T cells, residing in the colon and recognizing bacterial antigen presented by the MHC class II molecule.

The authors performed extensive characterization (phenotype, functional analyses in vitro and in vivo and transcriptomic analyses) of these cells from human samples and mouse model.

The authors have addressed the concerns raised during the first reviewing. They have strengthened the data supporting their claim on this T cell population they named TMIC.

It is an innovative and very interesting study shedding light on a subset of T cells that could participate to the development of inflammatory bowel disease.

Agnes Lehuen

Reviewer #3 (Remarks to the Author):

The paper by Hackstein et al has been substantially improved by revision and expanded with new analysis characterizing the CD161^{hi} CD4⁺ cell population in humans as well as a potentially similar population in mouse. This is an interesting story of interest for the field of mucosal immunology in health and disease.

My main concern remains: The naming of this population as T-MIC (MHC class II-restricted, Innate-like, Commensal-reactive) hinges primarily on the human data on figure 1 A-B where a very small proportion of the CD161^{hi} CD4 T cells are identified as reactive with these bacteria. So the extrapolation that all CD4 T cells with this phenotype is commensal-reactive is a stretch. Now, this claim is also supported by a mouse model, but still I am not very comfortable with the idea of giving this population a name based on a reactivity of less than 1% of that population in human.

There is a substantial background literature here which is hard to cover completely. However, I think one reference clearly missing is a study from Leslie Berg's lab identifying a similar PLZF⁺ CD4 T cell population in mice, which was described as commensal driven (Prince AL, J Immunol, 2014). This study I think should be cited and discussed.

RESPONSE TO REVIEWERS Round 2

We thank the reviewers for their input on our manuscript and hope the final version has satisfied their few remaining concerns.

Reviewer #1 (Remarks to the Author):

The authors have addressed my major concerns, given the experimental restraints.

We thank the reviewer for recognizing the experimental constraints and for their help improving the manuscript.

Reviewer #2 (Remarks to the Author):

This study describes of subset of T cells, residing in the colon and recognizing bacterial antigen presented by the MHC class II molecule.

The authors performed extensive characterization (phenotype, functional analyses in vitro and in vivo and transcriptomic analyses) of these cells from human samples and mouse model.

The authors have addressed the concerns raised during the first reviewing. They have strengthened the data supporting their claim on this T cell population they named TMIC.

It is an innovative and very interesting study shedding light on a subset of T cells that could participate to the development of inflammatory bowel disease.

Agnes Lehuen

We thank the reviewer for the positive remarks.

Reviewer #3 (Remarks to the Author):

The paper by Hackstein et al has been substantially improved by revision and expanded with new analysis characterizing the CD161hi CD4+ cell population in humans as well as a potentially similar population in mouse. This is an interesting story of interest for the field of mucosal immunology in health and disease.

My main concern remains: The naming of this population as T-MIC (MHC class II-restricted, Innate-like, Commensal-reactive) hinges primarily on the human data on figure 1 A-B where a very small proportion of the CD161hi CD4 T cells are identified as reactive with these bacteria. So the extrapolation that all CD4 T cells with this phenotype is commensal-reactive is a stretch. Now, this claim is also supported by a mouse model, but still I am not very comfortable with the idea of giving this population a name based on a reactivity of less than 1% of that population in human.

We appreciate the reservations of reviewer 3 in regard to naming new subsets. We have removed the name Tmic in the abstract but refer to the population by this name in the text for brevity. While we cannot be sure what proportion of commensal-reactive T cells in humans have this phenotype, we hope the reviewer is satisfied that this subset exists and may play an important role in the intestine based on the data presented.

There is a substantial background literature here which is hard to cover completely. However, I think one reference clearly missing is a study from Leslie Berg's lab identifying a similar PLZF+ CD4 T cell population in mice, which was described as commensal driven (Prince AL, J Immunol, 2014). This study I think should be cited and discussed.

We appreciate that there is a large literature to cover in regard to innate-like T cells and have included the citation suggested by this reviewer.